# Physiology-Aware Masked Cross-Modal Reconstruction for Biosignal Representation Learning

Hao Zhou [1 2]  Simon A. Lee [2]  Cyrus Tanade [2]  Keum San Chun [2]  Juhyeon Lee [2]  Migyeong Gwak [2]
Megha Thukral [2]  Justin Sung [2]  Eugene Hwang [3]  Mehrab Bin Morshed [2]  Li Zhu [2]  Viswam Nathan [2]
Md Mahbubur Rahman [2]  Subramaniam Venkatraman [2]  Sharanya Arcot Desai [2]

## Abstract

Biosignals acquired from different locations on the body often provide temporally ordered views of the same underlying physiological process. However, most existing self-supervised learning methods treat these signals as interchangeable views, overlooking the directional temporal dynamics that link them. A canonical example is the relationship between electrocardiography (ECG), which captures the electrical activation initiating each heartbeat, and photoplethysmography (PPG), which records the resulting peripheral pulse delayed by vascular dynamics. To capture this structured relationship, we introduce *xMAE*, a biosignal pretraining framework that leverages masked cross-modal reconstruction across temporally ordered biosignals as a training-time constraint to encourage physiologically meaningful timing structure in the learned representations. We show that pretraining with *xMAE* yields representations that outperform both unimodal and multimodal baselines on 15 of 19 downstream tasks, including cardiovascular outcome prediction, abnormal laboratory test detection, sleep staging, and demographic inference, while generalizing across devices, body locations, and acquisition settings. Further analysis suggests that the ECG–PPG timing structure is reflected in the learned PPG representations. Code is available at https://github.com/hzhou3/xMAE.

[1]The Pennsylvania State University [2]Samsung Research America [3]Samsung Electronics. Correspondence to: Hao Zhou <hao.zhou@psu.edu>, Sharanya Arcot Desai <s.desai1@samsung.com>.

*Proceedings of the 43rd International Conference on Machine Learning*, Seoul, South Korea. PMLR 306, 2026. Copyright 2026 by the author(s).

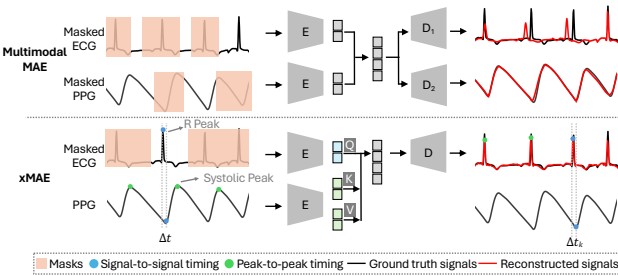

**Figure 1. Top:** Existing self-supervised approaches treat ECG and PPG as exchangeable views, overlooking their temporal relationship. **Bottom:** *xMAE* incorporates asymmetric masking and directional cross-attention to bias learning toward cross-modal temporal transition structure that reflects underlying cardiovascular dynamics relevant to health tasks.

## 1. Introduction

Biosignals frequently capture temporally ordered observations of the same underlying physiological process, a property we refer to as *asymmetric temporal observability*, which induces structured and directional relationships between modalities (Biagetti et al., 2018; Parchani et al., 2022). Photoplethysmography (PPG) and electrocardiography (ECG) provide a canonical example. ECG records the electrical activation that initiates each heartbeat, while PPG measures a delayed peripheral pulse shaped by vascular dynamics (Finnegan et al., 2023; Esmaelpoor et al., 2021). Many health-relevant changes manifest not only in waveform morphology, but also through variations in this temporal structure, such as pulse arrival time (Mukkamala et al., 2015; Block et al., 2020).

Despite its importance, most existing biosignal self-supervised learning approaches treat different modalities as interchangeable views, typically enforcing agreement through contrastive alignment (Chen et al., 2020) or joint reconstruction (Fang et al., 2024). This assumption obscures the directional and time-ordered nature of physiological sensing. We argue that temporal ordering and directionality constitute a strong inductive bias for learning biosignal representations that are both physiologically meaningful and broadly generalizable.

Motivated by this view, we frame biosignal representation learning as an inference problem under asymmetric temporal observability. We study how multimodal biosignals can serve as training-time scaffolding to improve representations of a single ubiquitous modality by encouraging representations to capture the transition structure, beyond unimodal waveform statistics.

**A General Framework** Building on this perspective, we introduce *xMAE*, a biosignal representation learning framework grounded in the principle: when modalities are temporally ordered, representation learning should respect the information flow. *xMAE* implements this principle through masked cross-modal reconstruction (Figure 1), biasing learning toward transition structures across modalities. In the ECG–PPG setting, the mask reveals the physiological structure where a heartbeat is electrically initiated and later observed as a peripheral pulse, while directional cross-attention encourages cross-modal learning rather than reliance on within-modality interpolation. We believe the framework is also applicable to other paired biosignals that observe different stages of an underlying process, such as ECG and ballistocardiography (Parchani et al., 2022) or muscle activation and motion (Biagetti et al., 2018).

**Robust PPG Representations** Pretrained on 9.4k hours of paired ECG–PPG recordings from MIMIC-III (Johnson et al., 2016), *xMAE* learns representations that consistently outperform strong unimodal and multimodal baselines across 15 out of 19 downstream tasks, including cardiovascular conditions, abnormal laboratory test, sleep staging, and demographic inference. These gains persist across datasets collected with different devices, body locations, and acquisition settings. Comparisons against open-source models further show that incorporating domain structure can rival the benefits of scaling data volume.

**Interpretability** We analyze ECG reconstruction behavior and cross-modal temporal alignment to probe what the model learns during pretraining. These analyses show that ECG–PPG timing structure emerges as an intrinsic property of the learned PPG representation space. As a result, the representations encode physiologically meaningful temporal dynamics that provide interpretable insight into cardiovascular structure relevant to downstream health tasks.

**Conflict of Interest Disclosure** Several authors are employees of Samsung Research America or Samsung Electronics, and this work was conducted in collaboration with Samsung Research America. The authors disclose these affiliations and related institutional support for transparency.

## 2. Related Work

Most foundation models for biosignals rely on generic self-supervised learning objectives originally developed for time series, vision, or audio (Dosovitskiy, 2020; He et al., 2022; Chen et al., 2020; Assran et al., 2022; Caron et al., 2021; Ansari et al., 2024; Dosovitskiy, 2020; Radford et al., 2021).

**Unimodal Models** These pretraining objectives are commonly applied in unimodal models that focus on a single physiological signal. Large-scale ECG models trained on clinical datasets have demonstrated strong performance on arrhythmia detection and related tasks (McKeen et al., 2025; Li et al., 2024). Yet, continuous ECG monitoring is impractical to deploy in daily life due to electrode requirements and user burden. In contrast, PPG-based foundation models have been explored to leverage the widespread availability of wrist-worn optical sensors and to incorporate waveform analysis into large-scale pretraining objectives (Pillai et al., 2024; Saha et al., 2025; Lee et al., 2025). While well-suited for passive monitoring and studied for robustness to noise such as motion artifacts (Ding et al., 2024), PPG remains a peripheral measurement that lacks the fine-grained electrical information from ECG (Allen, 2007).

**Multimodal Models** This challenge naturally led to multimodal models that jointly process multiple physiological signals (Thapa et al., 2026; Fang et al., 2024; Narayanswamy et al., 2024; Erturk et al., 2025). Common strategies include contrastive learning, where synchronized windows across modalities or views are pulled together while negative pairs are pushed apart (Chen et al., 2020; Nie et al., 2025), knowledge distillation (Caron et al., 2021), where a higher fidelity signal, such as ECG, guides the learning of a wearable signal representation, and masked autoencoders (MAE), where representations are learned through reconstruction of masked inputs (Fang et al., 2024; Narayanswamy et al., 2024) or through reconstruction of missing data in wearable streams (Xu et al., 2025). While these approaches leverage cross-signal consistency, they typically treat modalities as exchangeable views of the same underlying state. In cardiovascular sensing, this assumption may not always be valid, as ECG precedes PPG, and the temporal delays are relevant to vascular dynamics (Block et al., 2020).

**Summary** *xMAE* introduces an inductive bias that respects the directional relationship between electrical and mechanical cardiovascular signals. Prior work has explored generating ECG waveforms from PPG for data synthesis and augmentation. These approaches optimize for waveform realism, treating ECG generation as the end goal. In contrast, *xMAE* uses ECG reconstruction as a training-time scaffold rather than a target, leveraging it to inject physiologically grounded temporal structure into representation learning. As a result, our objective emphasizes transferable temporal abstractions rather than waveform fidelity, aligning pretraining with downstream health tasks that depend on timing dynamics rather than signal reconstruction. Further discussion is provided in Appendix G.5.

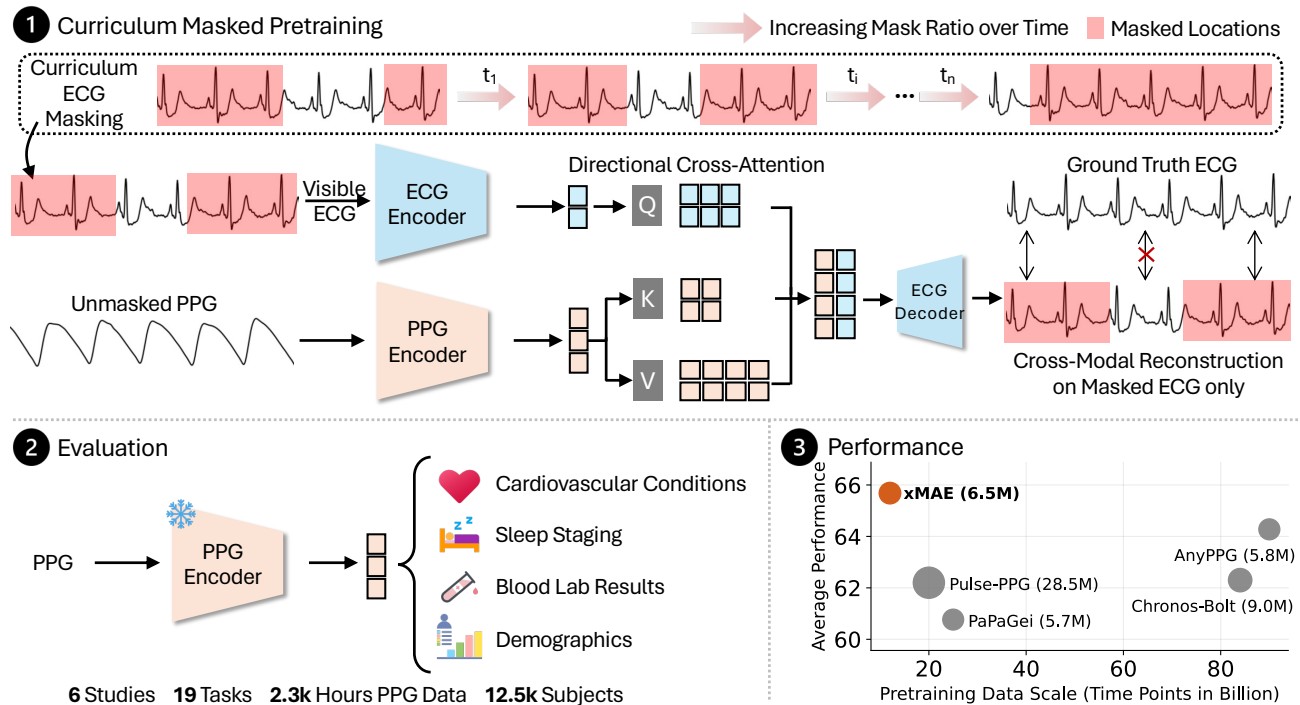

**Figure 2. Overview of *xMAE*.** (1) **Pretraining:** the model learns physiological structure by progressively reconstructing continuously masked ECG segments from synchronized PPG via directional cross-attention, encouraging the PPG encoder to capture underlying cardiac dynamics. (2) **Evaluation:** the PPG encoder is transferred to downstream tasks spanning cardiovascular conditions, sleep staging, blood lab results, and demographics across 6 studies (19 tasks; 2.3k hours of PPG; 12.5k subjects). (3) **Performance:** Despite a smaller pretraining data scale, *xMAE* achieves higher averaged classification performance compared to prior open-source foundation models.

## 3. Methodology

### 3.1. Pretraining under Exchangeable Views

Multimodal self-supervised learning methods (Fang et al., 2024; Thapa et al., 2026) traditionally assume temporal exchangeability, treating different modalities as synchronized views of a shared latent process. Under this formulation, contrastive alignment objectives encourage representations to emphasize modality-invariant features, implicitly assuming that temporal correspondence across signals is symmetric and instantaneous. Similarly, symmetric multimodal masked autoencoders reconstruct each modality from its surrounding temporal context, often allowing accurate reconstruction using within-modality interpolation alone when biosignals are locally smooth and highly predictable (Yu et al., 2006). As a result, the reconstruction objective can be satisfied without requiring the model to learn about cross-modal temporal relationships, motivating *xMAE*. We provide additional analysis and comparisons in Appendix G.

### 3.2. *xMAE*

**Definition & Overview**    We consider synchronized PPG (denoted as $P$) and ECG (denoted as $E$) signals collected from the same subject. Each input sample consists of a 10-second segment sampled at 100 Hz, yielding sequences $P \in \mathbb{R}^L$, $E \in \mathbb{R}^L$, $L = 1000$. Signal preprocessing details are provided in Appendix C.

Unlike prior works (Fang et al., 2024; Erturk et al., 2025), *xMAE* is proposed to reframe masked autoencoding as a structured inference problem that respects physiological relationships, with the masks designed to reflect domain knowledge about the direction of information flow between modalities. Formally, *xMAE* formulates a *cross-modal reconstruction* objective,

$$\hat{E}_{\mathcal{M}} = f_\theta(P, E_{\mathcal{V}}), \qquad (1)$$

where $\mathcal{M}$ denotes a masked subset of ECG time indices and $\mathcal{V}$ its complement. Under this formulation, ECG serves as a partially observed upstream signal, while PPG provides a delayed view of the underlying activity. During pretraining, PPG is always visible, whereas ECG is masked using *continuous temporal blocks* covering $M\%$ of the signal, with masking applied before encoding. We select $M\%$ to ensure at least one full cardiac cycle from ECG, and PPG remains visible, revealing physiologically meaningful cross-signal relationships such as timing (Block et al., 2020).

This design reconstructs ECG from PPG under asymmetric

masking, making precise temporal alignment a requirement for minimizing reconstruction error, biasing the model toward capturing information that is important under physiological transport, which is central to downstream health tasks (Mukkamala et al., 2015; Block et al., 2020). *xMAE* does not explicitly supervise peak locations; rather, physiologically meaningful structure emerges implicitly through directional reconstruction and temporal consistency.

**Curriculum ECG Masking Strategy** We adopt a curriculum masking strategy for ECG to progressively encourage cross-modal signal learning. Let $M \in (0, 1)$ denote the fraction of the ECG segment that is masked. Training begins with an initial masking ratio of $M_0 = 80\%$, and the masking ratio is increased in fixed steps of $5\%$ whenever the reconstruction loss improves by a predefined relative threshold, until reaching a maximum ratio of $90\%$, at which point at least one full cardiac cycle from both PPG and ECG remains visible. We justify this design choice in Appendix F.

**Signal Encoding** After masking, the input waveform to the encoder is $x \in \mathbb{R}^{L'}$, where $L' = L$ for PPG and $L' = |\mathcal{V}|$ for ECG. A modality-specific convolutional module then processes each continuous visible input signal while preserving temporal resolution and continuity, producing a feature map $x' \in \mathbb{R}^{C \times L'}$. The output is partitioned into non-overlapping temporal patches of length $P$ and linearly projected into $d$-dimensional token embeddings, yielding

$$Z \in \mathbb{R}^{N' \times d}, \quad N' = \left\lfloor \frac{L'}{P} \right\rfloor. \tag{2}$$

Learnable positional embeddings are added to encode temporal order. PPG tokens and visible ECG tokens are then embedded independently using Transformer encoders (Vaswani et al., 2017):

$$Z'_P = \text{Enc}_P(Z_P), \quad Z'_E = \text{Enc}_E(Z_E),$$

producing latent representations $Z'_P \in \mathbb{R}^{N \times d}$ and $Z'_E \in \mathbb{R}^{N_\mathcal{V} \times d}$. Each encoder operates only on visible inputs, following prior work (He et al., 2022).

**Directional Cross-Attention** To reconstruct the masked ECGs, masked ECG tokens are first reinserted using a shared learnable mask token and restored to their original temporal order, forming a full-length ECG token sequence, $\tilde{Z}_E$. We, then, employ a directional cross-attention mechanism in which ECG tokens act as queries and PPG tokens act as keys and values:

$$\text{Attn}(Q, K, V) = \text{softmax}\left(\frac{QK^\top}{\sqrt{d}}\right) V,$$
$$Q = \tilde{Z}_E, \quad K = Z'_P, \quad V = Z'_P. \tag{3}$$

Such attention is a standard operation, and is often used in Encoder-Decoder architecture and recent vision-language models (Vaswani et al., 2017; Alayrac et al., 2022); we utilize its directionality as an inductive bias to align with physiological structure, and to reflect the physiological dependency between modalities, encouraging the PPG encoder to capture physiologically relevant information.

**Cross-Modal Reconstruction Objective** A lightweight ECG decoder reconstructs the ECG waveform,

$$\hat{E} = \text{Dec}_E\big(\text{Attn}(\tilde{Z}_E, Z'_P, Z'_P)\big).$$

Pretraining minimizes mean squared error (MSE) over the masked locations on ECG:

$$\mathcal{L} = \mathbb{E}\left[\sum_{t \in \mathcal{M}} \|\hat{E}_t - E_t\|^2\right]. \tag{4}$$

## 4. Evaluation Setup

**Pretraining Dataset** We use the waveform matched subset of the MIMIC-III database (Johnson et al., 2016), which provides $\approx$3.4 million synchronized 10-second ECG and PPG recordings sampled at 100 Hz ($\approx$9.4k hours) collected in intensive care settings from $\approx$2.4k subjects after our preprocessing pipeline. This dataset enables large-scale self-supervised pretraining with high-quality paired physiological signals.

**Evaluation Datasets** We use public and institution-owned datasets collected in laboratory and free-living environments. These datasets, spanning across 6 studies from Samsung and DREAMT (Wang et al., 2025), include a wide range of cardiovascular, sleep, and demographic measurements, and reflect realistic deployment conditions for wearable health monitoring. Unless stated otherwise, we focus only on PPG signals because PPG can be collected passively and unobtrusively, making it well-suited for continuous health monitoring in daily-life settings. We provide a quick summarization of these datasets below (Table 2) with more details in Appendix E.1.

**Target 1** *Transferability of Learned PPG Representation*: We evaluate learned PPG representations on a diverse set of downstream tasks spanning both classification and regression, including cardiovascular risk (e.g., hypertension) assessment in both laboratory and free-living settings, arrhythmia-related event (e.g., premature ventricular contractions) detection, metabolic health indicators (e.g., Glycated Hemoglobin), demographic attribute (e.g., age) prediction, and sleep stage classification. Collectively, these tasks probe complementary aspects of cardiovascular health, hemodynamics, metabolic status, and sleep behavior.

A comprehensive set of baselines spanning unimodal and multimodal self-supervised learning approaches, as well as open-source foundation models, is included for comparison.

**Table 1.** Linear probing classification performance comparison against baselines on different tasks, including cardiovascular, labs, and sleep staging. The numeric values represent AUROC, and the standard deviation is reported in parentheses. The best performance is **bold**, the second best model is underscored. We conducted a t-test comparing *xMAE* (when it is the best) with the second-best model. $^*$ denotes $p < 0.05$, $^{**}$ denotes $p < 0.01$ and $^{***}$ denotes $p < 0.001$.

| Model | #param (M) | Cardiovascular Conditions | | | | Abnormal Blood Labs | | | | Sleep Staging | | | |
|---|---|---|---|---|---|---|---|---|---|---|---|---|---|
| | | Hyptn (lab) | Hyptn (free-living) | PVC | Ectopic Beats | A1C | Hemoglobin | Platelets | Sodium | Wake | Light | Deep | REM |
| MAE-1D | 6.7 | 53.6 (±4.5) | 55.1 (±0.9) | 78.7 (±5.0) | 85.8 (±2.3) | 48.5 (±8.6) | 53.1 (±13.5) | 62.7 (±13.2) | 58.2 (±9.9) | 64.6 (±1.7) | 56.1 (±1.5) | 55.5 (±3.0) | 51.5 (±1.5) |
| MSN | 6.5 | 56.1 (±5.4) | 54.6 (±0.7) | 68.5 (±4.3) | 69.2 (±1.3) | 51.6 (±7.1) | 60.0 (±14.6) | 67.9 (±15.3) | **63.1** (±12.1) | 63.1 (±1.7) | 55.7 (±1.1) | 52.2 (±4.0) | 52.4 (±1.6) |
| PaPaGei-P | 5.0 | 52.1 (±3.4) | 54.1 (±0.6) | 69.6 (±3.4) | 76.2 (±1.8) | 52.5 (±13.3) | 55.0 (±15.1) | 65.4 (±14.4) | 61.9 (±17.3) | 64.7 (±1.8) | 56.7 (±1.2) | **57.5** (±2.7) | 52.5 (±1.6) |
| Apple | 5.0 | 56.8 (±5.2) | 54.5 (±0.5) | 74.2 (±3.3) | 84.3 (±3.2) | 50.6 (±9.3) | 56.0 (±13.3) | 62.3 (±10.4) | 59.8 (±12.8) | 62.3 (±1.0) | 55.1 (±1.3) | 50.5 (±1.7) | 51.5 (±0.7) |
| DINO | 6.5 | 53.7 (±5.8) | 54.4 (±0.4) | 67.5 (±4.1) | 67.8 (±0.8) | 49.8 (±10.0) | 52.8 (±12.1) | 65.0 (±18.9) | 56.8 (±13.9) | 62.6 (±1.5) | 55.8 (±0.8) | 55.0 (±2.8) | 51.6 (±1.5) |
| LSM | 6.7 | 53.8 (±5.6) | 55.0 (±0.9) | 78.8 (±5.4) | 86.2 (±1.7) | 47.3 (±7.8) | 55.0 (±13.4) | 62.6 (±14.5) | 61.9 (±11.8) | 64.7 (±1.6) | 56.5 (±1.6) | **57.5** (±3.1) | 52.5 (±1.4) |
| SimCLR | 5.0 | 55.7 (±6.0) | 55.1 (±0.9) | 67.8 (±2.2) | 71.0 (±2.7) | 48.5 (±4.5) | 54.9 (±11.8) | 58.7 (±13.4) | 62.8 (±14.2) | 58.9 (±1.3) | 52.7 (±0.6) | 55.0 (±2.3) | 50.3 (±0.6) |
| Apple-M | 6.7 | 56.3 (±6.7) | 56.3 (±1.2) | 80.7 (±4.6) | 85.8 (±2.2) | 49.4 (±7.3) | 53.5 (±13.2) | **69.2** (±17.7) | 59.9 (±11.4) | 65.2 (±1.7) | 56.1 (±1.5) | 55.9 (±2.8) | 51.7 (±1.9) |
| *xMAE* | 6.5 | **68.8**$^{**}$ (±4.8) | **58.5** (±1.1)$^{***}$ | **81.4** (±5.1) | **87.8** (±2.3)$^{**}$ | **65.1** (±12.5)$^{**}$ | **62.0** (±16.0) | 68.6 (±16.5) | 61.7 (±16.1) | **66.4** (±2.3) | **57.5** (±1.3) | 55.9 (±5.3) | **54.5** (±2.5)$^*$ |

**Table 2.** Summary of datasets and downstream tasks.

| Dataset | # Subjects | # Segments | Setting | Task(s) |
|---|---|---|---|---|
| BP (lab) | 135 | 6,966 | Lab | Hypertension, BP regression |
| BP (free-living) | 9,427 | 28,344 | Free-living | Hypertension, BP regression |
| AFib | 139 | 480,717 | Lab | PVC detection |
| MX | 2,613 | 37,309 | Free-living | Ectopic beats detection |
| DREAMT | 100 | 235,419 | Sleep | Sleep staging (Wake/Light/Deep/REM) |
| Abnormal Lab Tests | 19+ | ~5k–13k | Free-living | A1C, Hemoglobin, Platelets, Sodium |

We provide details of these baselines in Appendix D.3 and pretraining protocols in Appendix D.4.

**Unimodal Baselines** We include baselines that only operate on PPG, including MAE-1D (He et al., 2022), MSN (Assran et al., 2022), PaPaGei-P (Pillai et al., 2024), and Apple (Abbaspourazad et al., 2023). We re-train these baselines with PPG signals.

**Multimodal Baselines** We also include multimodal baselines, such as DINO (Caron et al., 2021), LSM (Narayanswamy et al., 2024), SimCLR (Chen et al., 2020), and Apple-M (Fang et al., 2024). We re-train these baselines with PPG and ECG signals.

**Open-Weight Baselines** We further compare against physiological and time-series foundation models with publicly available weights, including PaPaGei (Pillai et al., 2024), Chronos-Bolt-Tiny (Ansari et al., 2024), AnyPPG (Nie et al., 2025), and Pulse-PPG (Saha et al., 2025). Although these models differ substantially in architecture, model size, pretraining data scale, and temporal resolution, we evaluate their released pretrained weights to provide a strong and representative reference point for assessing the impact of our proposed inductive bias.

**Protocol** Following prior works (Pillai et al., 2024; Narayanswamy et al., 2024), we report the area under the receiver operating characteristic curve (AUROC) for classification tasks, and mean absolute error (MAE) for regression tasks. All results are obtained using linear probing with 5-fold cross-validation split by subjects, where 20% of subjects are held out for testing in each fold. We chose this evaluation strategy because it provides a more comprehensive assessment of model performance across diverse users.

Performance is evaluated on the held-out subjects and is reported as the mean and standard deviation across five folds. We also conduct paired $t$-tests across folds against the strongest baseline and report statistical significance. Additional details of the evaluation setup are provided in Appendix E.2.

**Target 2** *Physiological Grounding Analysis*: We further show how the learned PPG representations are physiologically grounded through analyses of ECG reconstruction behavior, temporal alignment between the ground-truth and reconstructed ECG signals, and HRV features derived from the reconstructed ECG. More details are provided in Appendix G.3.

## 5. Evaluation Results

In this section, we first evaluate the transferability of PPG embeddings in §5.1. We then examine whether the learned representations are physiologically grounded through a case study on the physiological fidelity of reconstructed ECG signals in §5.2. Finally, we study key design choices and data efficiency in §5.3, and computational cost in §5.4.

### 5.1. Transferability of Learned PPG Representation

**Classification Evaluation** Table 1 reports linear probing performance on a diverse set of classification tasks, including hypertension classification in both laboratory and free-living settings, arrhythmia-related event detection (i.e., PVC and ectopic beats), metabolic health indicators such as A1C, and binary-class sleep staging. Out of 12 tasks, *xMAE* consistently outperforms unimodal or multimodal baselines across 9 tasks. The performance gains from 5 tasks are statistically significant. The performance gains are significantly pronounced (5 tasks), including hypertension (68.8 vs 56.8), ectopic beats (87.8 vs 86.2), and A1C (65.1 vs 52.5) when comparing against the strongest performing baseline. We believe, apart from waveform statistics, *xMAE* encodes features that are stable under physiological transport, leading to improved sensitivity to timing-related

**Table 3.** Linear probing regression performance comparison against baselines on different tasks, including blood pressure and demographics. The numeric values represent MAE, and the standard deviation is reported in parentheses. The best performance is **bold**, the second best model is underscored. We conducted a t-test comparing *xMAE* (when it is the best) with the second-best model. $^*$ denotes $p < 0.05$, $^{**}$ denotes $p < 0.01$ and $^{***}$ denotes $p < 0.001$.

| Model | #param (M) | BP Study (lab) | | BP Study (free-living) | | Demographics | | |
|---|---|---|---|---|---|---|---|---|
| | | Systolic BP | Diastolic BP | Systolic BP | Diastolic BP | Age (lab) | BMI (lab) | Age (free-living) |
| MAE-1D | 6.7 | 12.51 $(\pm1.19)$ | 9.39 $(\pm0.55)$ | 11.88 $(\pm0.28)$ | 9.47 $(\pm0.12)$ | 7.78 $(\pm1.16)$ | 3.86 $(\pm0.47)$ | 9.30 $(\pm0.21)$ |
| MSN | 6.5 | 12.41 $(\pm1.42)$ | 9.18 $(\pm0.87)$ | 11.82 $(\pm0.29)$ | 9.46 $(\pm0.13)$ | 7.45 $(\pm0.99)$ | 3.86 $(\pm0.49)$ | 9.45 $(\pm0.22)$ |
| PaPaGei-P | 5.0 | 12.95 $(\pm1.53)$ | 9.38 $(\pm1.02)$ | 11.82 $(\pm0.32)$ | 9.46 $(\pm0.14)$ | 7.60 $(\pm1.24)$ | **3.79** $(\pm0.50)$ | 9.70 $(\pm0.22)$ |
| Apple | 5.0 | 13.26 $(\pm1.88)$ | 9.32 $(\pm1.15)$ | 12.14 $(\pm0.30)$ | 9.65 $(\pm0.11)$ | 7.92 $(\pm1.23)$ | 4.01 $(\pm0.47)$ | 9.74 $(\pm0.19)$ |
| DINO | 6.5 | 12.86 $(\pm1.49)$ | 9.42 $(\pm1.02)$ | 11.86 $(\pm0.30)$ | 9.47 $(\pm0.12)$ | 8.13 $(\pm1.28)$ | 3.86 $(\pm0.46)$ | 9.63 $(\pm0.18)$ |
| LSM | 6.7 | 13.20 $(\pm1.84)$ | 9.35 $(\pm0.91)$ | 11.92 $(\pm0.29)$ | 9.49 $(\pm0.13)$ | 7.95 $(\pm1.13)$ | 3.86 $(\pm0.44)$ | 9.26 $(\pm0.19)$ |
| SimCLR | 5.0 | 13.88 $(\pm1.78)$ | 9.87 $(\pm1.11)$ | 12.83 $(\pm0.29)$ | 10.12 $(\pm0.13)$ | 8.48 $(\pm1.24)$ | 4.11 $(\pm0.44)$ | 10.68 $(\pm0.21)$ |
| Apple-M | 6.7 | 12.85 $(\pm1.61)$ | 9.07 $(\pm0.94)$ | 11.83 $(\pm0.31)$ | 9.47 $(\pm0.15)$ | 7.86 $(\pm1.14)$ | 3.89 $(\pm0.45)$ | 8.91 $(\pm0.20)$ |
| *xMAE* | 6.5 | **11.92** $(\pm1.42)$ | **8.65** $(\pm0.73)$ | **11.60** $(\pm0.31)^{***}$ | **9.30** $(\pm0.14)^{***}$ | **6.97** $(\pm1.14)^*$ | 4.09 $(\pm0.61)$ | **8.66** $(\pm0.20)^{***}$ |

**Table 4.** Performance comparison against open-source pretrained models on classification (AUROC) and regression (MAE) tasks using linear probing. *xMAE* achieves competitive or superior performance despite using fewer parameters and less pretraining data. We conducted a t-test comparing *xMAE* (when it is the best) with the second best model. $^*$ denotes $p < 0.05$, and $^{**}$ denotes $p < 0.01$.

| Model | Size | | | Classification (AUROC↑) | | | | Regression (MAE↓) | | | |
|---|---|---|---|---|---|---|---|---|---|---|---|
| | #param (M) | Hours (h) | Time Points (bil.) | Hypertension | Ectopic Beats | A1C | Wake | Systolic BP (free) | Diastolic BP (free) | Age (lab) | BMI (lab) |
| PaPaGei | 5.7 | 57k | 25 | 54.7 $(\pm3.9)$ | 80.9 $(\pm1.8)$ | 51.5 $(\pm9.1)$ | 64.7 $(\pm1.7)$ | 11.86 $(\pm0.29)$ | 9.49 $(\pm0.10)$ | 8.02 $(\pm1.38)$ | **3.79** $(\pm0.52)$ |
| AnyPPG | 5.8 | 100k×2 | 90 | 57.3 $(\pm1.2)$ | **89.3** $(\pm3.0)$ | **65.5** $(\pm15.8)$ | 65.1 $(\pm1.8)$ | 11.74 $(\pm0.31)$ | 9.41 $(\pm0.13)$ | 7.04 $(\pm1.17)$ | 4.01 $(\pm0.55)$ |
| Chronos-Bolt | 9.0 | - | 84 | 57.0 $(\pm0.9)$ | 85.7 $(\pm2.0)$ | 57.4 $(\pm12.4)$ | 65.1 $(\pm1.7)$ | 11.75 $(\pm0.29)$ | 9.42 $(\pm0.15)$ | 7.65 $(\pm1.28)$ | 3.95 $(\pm0.52)$ |
| Pulse-PPG | 28.5 | 55k | 20 | 57.8 $(\pm0.9)$ | 83.9 $(\pm1.9)$ | 59.9 $(\pm12.9)$ | 64.5 $(\pm2.7)$ | 11.73 $(\pm0.31)$ | 9.36 $(\pm0.14)$ | 7.15 $(\pm0.50)$ | 4.27 $(\pm0.55)$ |
| *xMAE* | 6.5 | 9.4k×2 | 6 | **58.5** $(\pm1.1)$ | 87.8 $(\pm2.3)$ | 65.1 $(\pm12.5)$ | **66.4** $(\pm2.3)$ | **11.60** $(\pm0.31)^{**}$ | **9.30** $(\pm0.14)^{**}$ | **6.97** $(\pm1.14)$ | 4.09 $(\pm0.61)$ |

patterns associated with cardiovascular conditions. The consistent improvements across heterogeneous classification tasks further suggest that the learned representations generalize beyond a single clinical endpoint, providing a robust foundation for a wide range of wearable health applications and underscoring the impact of our proposed framework.

**Regression Evaluation** Table 3 reports linear probing performance on a range of regression tasks, including systolic and diastolic blood pressure estimation as well as demographic attribute prediction, evaluated in both laboratory-controlled and free-living environments. Across 6 out of 7 tasks, *xMAE* consistently achieves lower errors than unimodal PPG-only baselines and multimodal self-supervised methods, indicating stronger generalization to continuous-valued physiological outcomes. This suggests that our learned representations are robust to real-world variability and capture stable, underlying cardiovascular dynamics.

**Open-weight Baseline Evaluation** Table 4 compares *xMAE* against recent open-source physiological or time-series foundation models with varying model sizes, training durations, and temporal resolutions. Despite differences in pretraining scale and architecture, *xMAE* demonstrates competitive or superior performance across downstream tasks (More results are provided in Appendix H.1). These results suggest that our pretraining objective can be as important as increasing model scale (Pulse-PPG) or training data volume

(PaPaGei, Chronos-Bolt, or AnyPPG). While health applications could benefit from multimodal pretraining (AnyPPG and *xMAE*), building robust biosignal models with reduced reliance on paired data and more efficient use of limited supervision remains future work.

**Comparing with Hand-crafted Features** We extract 35 handcrafted PPG features, including amplitude, morphology, derivative, and timing features based on Neurokit2 (Makowski et al., 2021). We train a logistic regression classifier on top of these features for different tasks. Table 5 shows that handcrafted PPG features remain competitive on some tasks, especially in the free-living hypertension task where the gap is small. However, *xMAE* improves over both the full set and the timing-only features across all tasks. This suggests that *xMAE* captures information beyond standard PPG feature engineering, including broader physiological structure that may not be fully represented by manually designed features.

**Table 5.** Comparison with feature-engineering baselines. Results are reported as AUROC, where higher is better.

| Task | All Features | Timing-Only Features | *xMAE* |
|---|---|---|---|
| Hyptn (lab) | 59.7 $(\pm13.8)$ | 53.1 $(\pm10.6)$ | 68.8 $(\pm4.8)$ |
| Hyptn (free-living) | 58.4 $(\pm1.7)$ | 57.7 $(\pm1.2)$ | 58.5 $(\pm1.1)$ |
| PVC | 76.7 $(\pm4.9)$ | 72.2 $(\pm3.6)$ | 81.4 $(\pm5.1)$ |
| Ectopic Beats | 83.5 $(\pm2.2)$ | 83.8 $(\pm1.3)$ | 87.8 $(\pm2.3)$ |
| A1C | 58.9 $(\pm21.3)$ | 35.3 $(\pm12.1)$ | 65.1 $(\pm12.5)$ |
| Wake | 63.7 $(\pm1.2)$ | 62.6 $(\pm1.9)$ | 66.4 $(\pm2.3)$ |

**Other Results** While our main focus is to evaluate the transferability of PPG embeddings via linear-probing, we also include fine-tuning results in Appendix H.4.

**Summary** Improvements achieved by *xMAE* are statistically significant in most settings when compared to the strongest baselines. These results demonstrate that our pretraining leads to more informative and generalizable representations for biosignal health applications. Furthermore, the observed gains suggest that modeling cross-modal temporal transitions can be as important as increasing model scale or expanding pretraining data volume, highlighting the value of promoting domain-specific structure into biosignal representation learning.

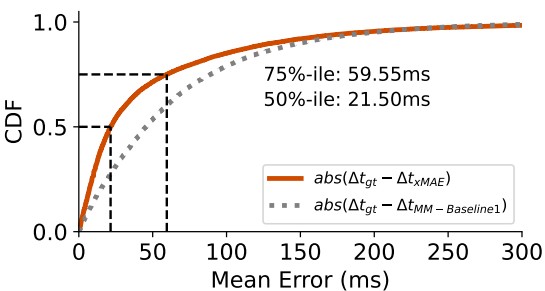

**Figure 3. Delay Comparison.** Cumulative distribution functions (CDFs) of absolute ECG–PPG delay error, measured as the difference between the ground-truth delay ($\Delta t_{gt}$) computed from paired ECG–PPG signals and the delay estimated from reconstructed signals. The delay is approximated as the temporal offset between the ECG R-peak and the PPG onset valley. *xMAE* exhibits lower delay error compared to a multimodal MAE baseline, with a median error of 21.5ms, indicating improved preservation of physiologically meaningful timing relationships.

## 5.2. Physiological Grounding Analysis

***xMAE* is Physiologically Grounded** To validate our claim that the learned PPG representations in *xMAE* encode physiologically meaningful ECG–PPG timing relationships, we compare against a masked autoencoding baseline that jointly encodes ECG and PPG and reconstructs both modalities (Fang et al., 2024; Narayanswamy et al., 2024). We quantify this timing as the temporal difference between the ECG R-peak and the PPG onset valley for both *xMAE* and the baseline. Figure 3 evaluates how well *xMAE* preserves this physiological delay by measuring the absolute error between the ground-truth delay, $\Delta t_{gt}$, computed from ground truth ECG–PPG signals, and the delay computed from reconstructed ECG and paired PPG. *xMAE* achieves a median delay error of 21.5ms, corresponding to a 53.3% reduction relative to the multimodal MAE baseline. This improvement is consistent across the error distribution, indicating more accurate preservation of beat-level ECG–PPG timing relationships. These results support that *xMAE*, given PPG waveforms, learns to reason physiologically meaningful re-

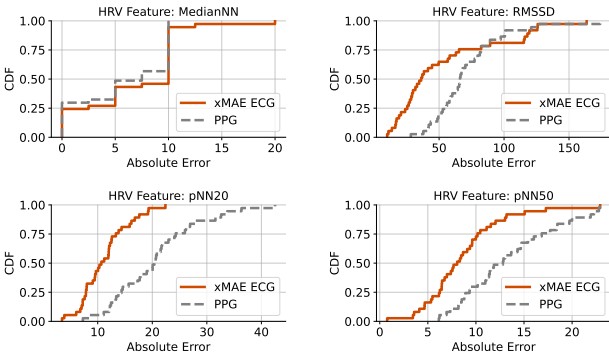

**Figure 4. Evaluating ECG reconstruction quality via HRV features.** CDFs of absolute error for HRV metrics computed from *xMAE*-reconstructed ECG and from PPG signals. Across all features, *xMAE* exhibits consistently lower error compared to PPG based HRV feature calculation with NeuroKit2 (Makowski et al., 2021), a state-of-the-art Python toolbox for neurophysiological signal processing. Overall, the beat-to-beat timing and temporal structure of ECG–PPG is well-preserved in *xMAE*.

lationships between PPG and ECG, which is important for capturing cardiovascular dynamics that are critical for robust health monitoring. Additional details with visualizations (Figure 15 and Figure 16) are provided in Appendix G.3.

**Case Study: Physiological Fidelity of Reconstructed ECG** We assess ECG reconstruction quality using heart rate variability (HRV) features, including MedianNN, RMSSD, pNN20, and pNN50, which are widely used in applications such as sleep analysis, stress monitoring, and cardiovascular risk assessment (Shaffer & Ginsberg, 2017). Figure 4 depicts that, across these HRV metrics, *xMAE*-reconstructed ECGs exhibit lower error compared to using PPG alone. MedianNN primarily captures longer-timescale variability in heart rate over the recording window, making it more sensitive to slow trends, baseline drift, and low-frequency noise. In contrast, RMSSD and pNN metrics emphasize short-term, beat-to-beat variability, focusing on rapid fluctuations between successive heartbeats (Shaffer & Ginsberg, 2017). Overall, these results demonstrate that *xMAE* encodes meaningful ECG dynamics in its latent PPG space. More details of the reconstruction procedure, evaluation details, and visualizations (Figure 20 and Figure 21) are provided in Appendix J.

## 5.3. Ablation Study and Data Efficiency

**Ablation Study** We also evaluate the effectiveness of design choices in *xMAE*, as summarized in Table 6. Continuous ECG masking consistently outperforms random masking, demonstrating that removing continuous temporal segments is crucial for preventing trivial local interpolation and promoting cross-signal learning. Utilizing curriculum masking yields improved performance, indicating

**Table 6. Design choices.** The default settings are in gray.

|  | Hypertension (lab) | Ectopic Beats |
|---|---|---|
| Random | 63.0 (±8.2) | 84.4 (±3.1) |
| Continuous | **68.8** (±4.8) | **87.8** (±2.3) |

**(a) Mask Type**

|  | Hypertension (lab) | Ectopic Beats |
|---|---|---|
| Fixed Ratio | 66.9 (±7.3) | 85.8 (±2.4) |
| w/ Curriculum | **68.8** (±4.8) | **87.8** (±2.3) |

**(b) Strategy of Mask Ratio**

|  | Hypertension (lab) | Ectopic Beats |
|---|---|---|
| Multi-Recons. | 65.8 (±5.5) | 83.4 (±2.1) |
| Cross-Recons. | **68.8** (±4.8) | **87.8** (±2.3) |

**(c) Training Loss**

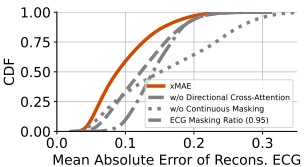 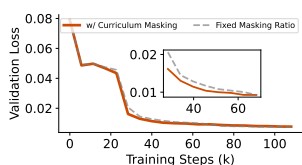

**Figure 5. Ablation study. Left:** CDFs of mean absolute reconstruction error for ECG under different ablated variants of *xMAE*. **Right:** Validation loss curves during pretraining comparing curriculum masking with a fixed masking ratio (90%).

that masking progressively enforces reliance on PPG while retaining sufficient ECG context. Replacing multimodal self-reconstruction with the proposed cross-reconstruction objective leads to substantial gains. Figure 5 (Left) reports the reconstructed errors of ECG when designs in *xMAE* were replaced. In short, they collectively contribute to the performance. We provide more details and visualizations in Appendix I. We also compare the validation loss of pretraining from our curriculum masking and a fixed high masking ratio (90%) in Figure 5 (Right). The curriculum-based curve exhibits a stable loss decrease in early and mid-stage, providing a principled mechanism to align the pretraining with our intended cross-modal reasoning objective. An additional justification is provided in Appendix F.

**Data Efficiency**    Figure 6 evaluates the data efficiency of *xMAE* under both reduced pretraining data and limited labeled finetuning. In the pretraining data volume study, *xMAE* consistently outperforms multimodal baselines on ectopic beat detection across all data scales, with particularly strong gains when only a small fraction of pretraining data is available, suggesting that our framework enables more effective utilization of limited unlabeled data. In the few-shot finetuning setting, *xMAE* achieves substantial improvements over baselines when only a small number of labeled PPG segments per subject are provided and maintains strong per-

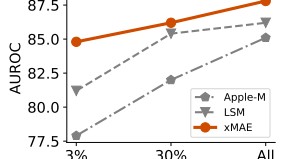 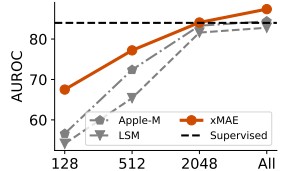

**Figure 6. Pretraining Data Volume (Left):** Linear probing performance on *Ectopic Beats* under varying pretraining data volume, where *xMAE* consistently outperforms baseline methods across all data scales. **Few-Shot Finetuning (Right):** Performance on *PVC* detection with varying numbers of labeled PPG segments per patient, where *xMAE* exhibits strong gains in low-label regimes and maintains strong performance as supervision increases.

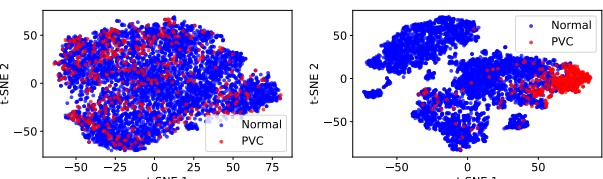

**Figure 7. Visualization of finetuned PPG embedding spaces for PVC classification.** Two-dimensional t-SNE projections of finetuned PPG embeddings learned by PaPaGei-P (left) and *xMAE* (right), colored by ground truth class labels (Normal vs. PVC).

formance as supervision increases. Together, these results demonstrate that representations learned by *xMAE* are both data- and label-efficient, making it well-suited for practical wearable health applications where large-scale annotation is costly or infeasible. Figure 7 demonstrates finetuned PPG embeddings of *xMAE* and PaPaGei-P (we pretrain with the same data as *xMAE*) from the PVC classification task. Notably, embeddings produced by *xMAE* show more distinct class structure in the representation space, suggesting that *xMAE* learns PPG embeddings consistent with its stronger downstream performance. Additional results and visualizations are provided in Appendix H.3 and Appendix K.

### 5.4. Computational Cost Analysis

We compare the computational cost of *xMAE* against representative open-source biosignal foundation models, including PulsePPG, AnyPPG, and PaPaGei. We report three complementary metrics: the number of trainable parameters, theoretical computational complexity measured in GFLOPs, and empirical inference throughput measured in segments per second. All experiments are performed on an NVIDIA H200. As shown in Table 7, *xMAE* achieves a strong efficiency–performance trade-off, delivering competitive computational efficiency while retaining sufficient capacity for robust downstream transfer. This gives *xMAE* great potential for large-scale pretraining and deployment on resource-constrained wearables. A full analysis is provided in Appendix L.

**Table 7.** Computational cost comparisons.

| Model | # Params (M) | GFLOPs | Throughput (k segments/sec) |
|---|---|---|---|
| PulsePPG | 28.5 | 28.5 | 3.4 |
| AnyPPG | 5.8 | 0.194 | 22 |
| PaPaGei | 5.7 | 0.0598 | 33 |
| *xMAE* | 6.5 | 0.165 | 24 |

## 6. Discussion and Limitation

**A General Self-Supervised Learning Framework under Asymmetric Temporal Observability**    Beyond ECG and PPG, *xMAE* represents a general self-supervised learning framework for paired signals that observe structured, temporally ordered stages of the same underlying process. We believe it is naturally applicable to settings such as ECG–ballistocardiography (Parchani et al., 2022), EEG–fNIRS (Ahn & Jun, 2017), EMG–motion (Biagetti et al., 2018), and combinations of inertial and biosignals (Zhou et al., 2025). By adopting masked asymmetric cross-modal reconstruction as the pretraining objective, *xMAE* provides a simple yet principled mechanism for injecting domain structure into representation learning, without relying on black-box waveform modeling. This perspective is particularly timely given the recent surge of interest in health-focused foundation models and large-scale self-supervised pretraining for clinical and wearable data. As health AI systems are increasingly deployed in high-stakes settings, it becomes critical that pretraining objectives reflect how physiological processes unfold over time, rather than treating biosignals as generic sequences. By preserving interpretable temporal structure, such as cross-modal timing relationships, *xMAE* supports more transparent and physiologically grounded representation learning, helping pave the way toward trustworthy and robust foundation models for health applications (Ahmad et al., 2018; Choi et al., 2016).

**Generalization from Lab to Wearables**    During pretraining, ECG provides a precise temporal reference that aligns each heartbeat with the onset of the corresponding PPG pulse, encouraging *xMAE* to organize PPG representations around beat-level timing structure rather than device-specific waveform characteristics. As a result, the model learns how temporal information is expressed intrinsically in the PPG signal itself. Although clinical and wearable signals differ in sensor placement, signal quality, and motion noise, our matched-domain analysis shows that pretraining on wrist PPG leads to similar downstream performance as pretraining on fingertip PPG (Appendix H.2). This suggests that the learned ECG–PPG structure is not tied to a single sensing location, but transfers across the fingertip-to-wrist distribution shift. Our observed cross-device and cross-subject transferability is consistent with prior work (Saha et al., 2025; Pillai et al., 2024), which shows that representations pretrained on clinical datasets can generalize effectively to wearable data, achieving performance comparable to models pretrained in the reverse direction. Together, these findings suggest that the temporal structure is conserved across devices and subjects. More broadly, *xMAE* enables biosignal representation learning to leverage diverse and heterogeneous data sources beyond consumer devices.

**Limitations**    We acknowledge several limitations.

- The current pretraining procedure requires paired ECG and PPG data, which may not always be available at scale. Future work could reduce this reliance by more efficiently leveraging limited supervision. Nevertheless, as shown in §5, pretraining on a moderate amount of clinically collected paired data still yields transferable PPG representations that generalize across tasks, hardware, and acquisition settings.

- We use a MSE loss, which is simple yet effective in practice and captures dominant ECG R peak characteristics, but may not model finer-grained ECG timing information, such as P–R intervals. Exploring supervision that incorporates even richer timing features may further enhance temporal sensitivity and is left for future work.

- Although our analysis provides evidence that *xMAE* preserves ECG features such as R peaks, the interpretability of the latent space remains limited. The current analysis focuses on reconstructed ECG morphology and does not fully reveal how individual latent dimensions correspond to specific physiological mechanisms. Future work could develop more direct probing, attribution, and concept-based analyses to better understand what physiological factors are encoded in the learned representation.

**Ethics**    This work uses de-identified physiological data from MIMIC-III, a publicly available clinical dataset released under established data use agreements and institutional review processes. In addition, we evaluate our approach on several wearable datasets collected under institutional review board approval with informed consent from participants; all data were de-identified prior to analysis. We do not claim direct clinical decision-making capability, and models trained with this framework should be evaluated carefully for bias, robustness, and safety before any health-related deployment. We acknowledge that the MIMIC-III population is demographically skewed, with a large fraction of White and older ICU patients. This may limit generalizability to underrepresented demographic groups. Because not all downstream datasets provide complete demographic labels, we do not make claims about demographic fairness. Future studies should evaluate subgroup performance across race, sex, age, and skin tone before deployment in health-related applications.

## Impact Statement

This work advances self-supervised representation learning by demonstrating that incorporating structured inductive biases into pretraining can be more effective than relying on data scale alone. By framing multimodal learning as an inference problem with directional and temporal constraints, our approach shows how limited paired data can be leveraged to learn transferable representations from peripheral signals. In health and biomedical settings, this perspective supports more interpretable and data-efficient learning from passive and widely available measurements. Beyond biosignals, these results highlight a general strategy for representation learning in settings where modalities observe different, temporally ordered stages of an underlying process, offering a principled alternative to exchangeable-view assumptions commonly used in multimodal pretraining.

## Acknowledgments

We thank Praveen Raja, Matthew Wiggins, Mike Freedmanm Esther Song, and Jacob Kim for their behind-the-scenes support, which helped make the project run smoothly.

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

# A. Author Contribution Breakdown

We attribute proper credit to the following authors for their contributions in this project.

**Table 8.** Overview of author contributions.

| Author | Concept | Experiment Design | Coding | Analysis | Writing | Visualization | Project Mgmt. | Discussion | Resources |
|---|---|---|---|---|---|---|---|---|---|
| Hao Zhou | ✓ | ✓ | ✓ | ✓ | ✓ | ✓ | ✓ | ✓ | |
| Simon A. Lee | | ✓ | | ✓ | ✓ | | | ✓ | |
| Cyrus Tanade | | | | ✓ | ✓ | | | ✓ | ✓ |
| Keum San Chun | | | | | | | | ✓ | ✓ |
| Juhyeon Lee | | | | | | | | ✓ | |
| Migyeok Gwak | | | | | | | | ✓ | ✓ |
| Megha Thurkal | | | | | ✓ | | | ✓ | |
| Justin Sung | | | | | | | | ✓ | |
| Eugene Hwang | | | | | | | | ✓ | ✓ |
| Mehrab Bin Morshed | | | | | | | | ✓ | |
| Li Zhu | | | | | | | | ✓ | |
| Viswam Nathan | | | | | | | | ✓ | |
| Md Mahbubur Rahman | | | | | | | | ✓ | ✓ |
| Subramaniam Venkatraman | | | | | | | ✓ | ✓ | ✓ |
| Sharanya Acot Desai | | ✓ | | ✓ | ✓ | ✓ | ✓ | ✓ | ✓ |

# B. LLM Usage

We utilize a large language model (LLM) to improve the clarity and readability of the text based on author-provided drafts. All scientific content, experimental design, and analysis were conceived, implemented, and verified by the authors.

# C. Signal Preprocessing Pipeline

To facilitate pretraining and evaluation, we follow a standard preprocessing pipeline that ensures high-quality PPG and ECG segments. This preprocessing pipeline for PPG and ECG is consistent across all pretraining and evaluation studies. Next, we detail these steps as follows.

**PPG Preprocessing**    First, given a full sequence of PPG signal, we perform a Butterworth bandpass filtering with a low cut frequency of 0.5 Hz, a high cut frequency of 8 Hz, and an order of 3 (Christiano & Fitzgerald, 2003). Then, we take 10-second windows from this long sequence without overlapping, and perform a quality check by utilizing the function, `ppg_quality`, in Neurokit2 (v0.2.12) (Makowski et al., 2021) with the method being `templatematch`. Note that this function will return an array of quality scores ([0, 1]), and we only keep the segments that have the 15%-ile score larger than 0.9. Finally, we normalize the high-quality segments to the range [−1, 1] for stability during training and evaluation.

**ECG Preprocessing**    First, given a full sequence of ECG signal, we perform a highpass filtering with a low cut frequency of 0.5 Hz, and an order of 5 (Christiano & Fitzgerald, 2003), followed by a powerline filtering to filter out 50 Hz powerline noise and smooth the signal with a moving average kernel with the width of one period of 50Hz. Then, we take 10-second windows from this long sequence without overlapping (we ensure the windows are aligned with paired PPG windows). Then, similar to PPG quality filtering, we utilize the function, `ecg_quality`, in Neurokit2 (v0.2.12) (Makowski et al., 2021) with the method being `templatematch`. Note that this function will return an array of quality scores ([0, 1]), and we only keep the segments that have the 15%-ile score larger than 0.9. Finally, we normalize the high-quality segments to the range [−1, 1] for stability during training and evaluation.

Both PPG and ECG signals are resampled to 100 Hz during pretraining or evaluation.

# D. Pretraining Dataset, Baselines, Protocols

In this section, we provide details of the pretraining dataset, split, baselines, and training protocols.

### D.1. Pretraining Dataset and Split

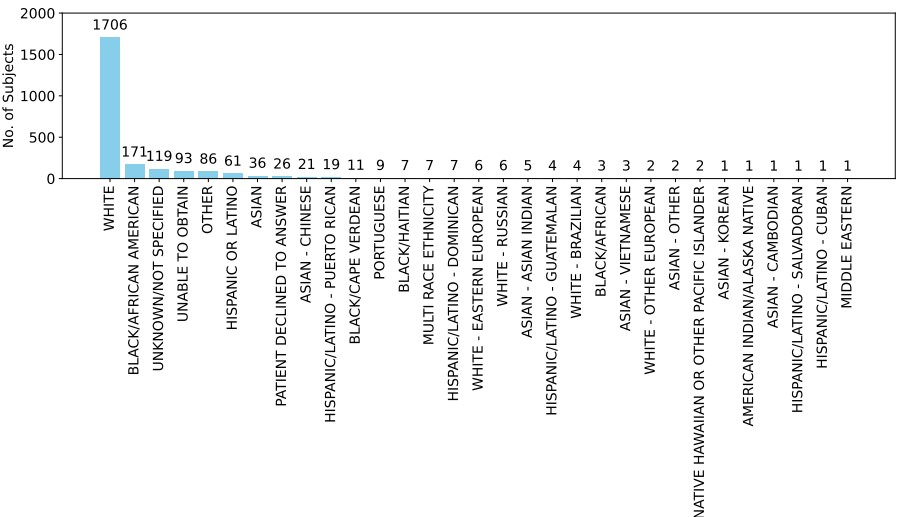

**Figure 8.** Ethnic group distribution of the subjects in the pretraining dataset.

We mainly use the waveform matched subset of the MIMIC-III database (Johnson et al., 2016), which provides 3.4M synchronized 10-second ECG and PPG recordings sampled at 100 Hz (9.4k hours) collected in intensive care settings from 2.4k subjects (Figure 8 depicts the ethnic group distribution) after our preprocessing pipeline. This dataset enables large-scale self-supervised pretraining with high-quality paired physiological signals. While there are other datasets with paired PPG and ECG, we are limited to include under our industrial settings. We plan to increase the data volume and explore how to build strong physiologically grounded foundation models with reduced reliance on paired multimodal data and more efficient use of limited supervision. Nevertheless, we believe the current settings have demonstrated the value of promoting domain-specific knowledge into biosignal pretraining.

We split the dataset into two parts by subjects, resulting in 3.1M segments from 2.1k subjects (90%) in the pretraining set, and the remaining 10% is used for validation to prevent overfitting.

### D.2. Model Architecture: *xMAE*

This appendix provides a complete architectural specification of *xMAE*, corresponding to the hyperparameters summarized in Table 9.

**Input**    We consider paired photoplethysmography (PPG) and electrocardiography (ECG) signals collected synchronously from the same subject. Each input sample consists of a 10-second segment sampled at 100 Hz, yielding sequences

$$P \in \mathbb{R}^L, \qquad E \in \mathbb{R}^L, \qquad L = 1000. \tag{5}$$

**Curriculum ECG Masking Strategy**    We adopt a curriculum learning strategy over the ECG masking ratio to progressively encourage cross-modal reasoning. Let $M \in (0, 1)$ denote the fraction of the ECG segment that is masked. Training begins with an initial masking ratio of $M_0 = 80\%$, and the masking ratio is increased in fixed steps of $5\%$ whenever the reconstruction loss improves by a predefined relative threshold (10% in our implementation), until reaching a maximum masking ratio of $M_{\max} = 90\%$, at which point at least one full cardiac cycle from both PPG and ECG remains visible, revealing physiologically meaningful cross-signal relationships such as timing (Block et al., 2020). At lower masking stages, a non-trivial portion of the ECG signal is visible, providing informative temporal and morphological cues that guide early training and stabilize optimization. As the masking ratio increases, the available ECG context becomes progressively more limited, forcing the model to rely more heavily on PPG signals for reconstruction.

**Signal Encoding**    After masking, the input waveform to each encoder is represented as

$$x \in \mathbb{R}^{L'}, \tag{6}$$

where $L' = L$ for PPG and $L' = |\mathcal{V}|$ for ECG.

Each modality is processed by an independent convolutional module that preserves temporal resolution and continuity and outputs a feature map

$$x' \in \mathbb{R}^{C \times L'}, \qquad C = 32. \tag{7}$$

The feature map is partitioned into non-overlapping temporal patches of length $P = 40$, which are linearly projected into a $D$-dimensional embedding space with $d = 256$. This yields

$$Z \in \mathbb{R}^{N' \times d}, \qquad N' = \left\lfloor \frac{L'}{P} \right\rfloor. \tag{8}$$

For fully observed PPG, this results in $N = 25$ tokens per segment (length is 40; $40 \times 25 = 1000$). Learnable positional embeddings are added to encode temporal order. PPG and visible ECG tokens are then processed independently by modality-specific Transformer encoders:

$$Z'_P = \text{Enc}_P(Z_P), \qquad Z'_E = \text{Enc}_E(Z_E). \tag{9}$$

The PPG encoder consists of two Transformer blocks, while the ECG encoder uses one block. All blocks use 8 attention heads, embedding dimension 256, and dropout rate 0.1. Each encoder operates only on visible tokens.

**Directional Cross-Attention**    To reconstruct the masked ECG segment, masked ECG tokens are first reinserted using a shared learnable mask token and restored to their original temporal order, forming a full-length ECG token sequence, $\tilde{Z}_E$. We, then, employ a directional cross-attention mechanism in which ECG tokens act as queries and PPG tokens act as keys and values:

$$\begin{aligned} \text{Attn}(Q, K, V) &= \text{softmax}\left(\frac{QK^\top}{\sqrt{d}}\right) V, \\ Q &= \tilde{Z}_E, \quad K = Z'_P, \quad V = Z'_P. \end{aligned} \tag{10}$$

This design reflects the physiological dependency between modalities: given partial electrical activity from ECG, the model queries PPG to retrieve temporally relevant hemodynamic information. The cross-modal bridge consists of a single cross-attention block and enables sample-specific temporal alignment without assuming a fixed ECG–PPG delay.

**Reconstruction Objective**    A lightweight ECG decoder consisting of a single Transformer block maps the fused representations back to the waveform domain:

$$\hat{E} = \text{Dec}_E\big(\text{Attn}(\tilde{Z}_E, Z'_P, Z'_P)\big). \tag{11}$$

Training minimizes mean squared error (MSE) over the masked ECG interval:

$$\mathcal{L} = \mathbb{E}\left[ \sum_{t \in \mathcal{M}} \|\hat{E}_t - E_t\|^2 \right]. \tag{12}$$

By restricting supervision to the masked region, the loss penalizes both amplitude errors and temporal misalignment, encouraging the model to learn physiologically grounded electrical-to-mechanical timing relationships.

**# Parameters**    We provide the detailed number of trainable parameters. In particular, our *PPG module* has 2.85M parameters, making it suitable for on-device inference (Tan & Le, 2019). Despite not being used for downstream tasks evaluation, our *ECG module* and the *Directional Cross-Attention module* have 2.06M and 1.58M parameters, respectively.

**Computational Costs**    We provide an analysis of computational costs in Appendix L.

### D.3. Baselines

**MAE-1D** (He et al., 2022)    MAE-1D extends the masked autoencoding paradigm to one-dimensional time-series signals. The baseline employs a transformer-based encoder trained to reconstruct masked temporal patches from partially observed inputs using random masking. By learning contextual representations over long temporal windows, MAE-1D captures

**Table 9.** Default architectural hyperparameters of *xMAE*.

| Hyperparameter | Value |
|---|---|
| Input sequence length ($L$) | 1000 |
| Patch length ($P$) | 40 |
| Number of patches ($N = L/P$) | 25 |
| Conv Module output channels ($C$) | 32 |
| Conv Module kernel size | 3 |
| Conv Module channel widths | (32,64,128) |
| Token embedding dimension ($D$) | 256 |
| Projection dimension | 384 |
| Transformer heads | 8 |
| PPG encoder depth | 2 blocks |
| ECG encoder depth | 1 block |
| Bridge depth | 1 cross-attention block |
| ECG decoder depth | 1 block |
| Dropout | 0.1 |
| ECG Masking Ratio | 80% → 90% |

generic structure in sequential time series and has been shown to transfer effectively across diverse downstream time-series tasks. In our experiments, we adopt MAE-1D as a unimodal self-supervised baseline and apply it to PPG signals.

**MSN** (Assran et al., 2022)   Masked Siamese Networks (MSN) aim to learn label-efficient representations by integrating masked signal modeling with Siamese-style self-supervised objectives. The method masks portions of the input signal and encourages agreement between multiple augmented views, eliminating the need for explicit class labels. MSN adopts a Vision Transformer encoder shared across views and incorporates a lightweight network to stabilize training. By coupling self-distillation with masked reconstruction, MSN reduces sample complexity and improves representation learning under limited supervision. In our experiments, we adopt MSN as a unimodal self-supervised baseline applied to PPG signals.

**PaPaGei-P** (Pillai et al., 2024)   PaPaGei is a domain-specific foundation model tailored for optical physiological sensing, with a particular focus on PPG. The approach employs a ResNet-style convolutional architecture to learn robust and generalizable representations from large-scale optical physiological datasets. For this baseline, we adopt the official PaPaGei implementation and follow its participant-level pretraining strategy, retraining the model on our employed dataset to ensure a fair and controlled comparison.

**Apple** (Abbaspourazad et al., 2023)   This model introduces a self-supervised learning objective for wearable physiological signals, with a particular focus on large-scale PPG or ECG data. Rather than proposing a new architecture, researchers from APPLE contribute a loss function that encourages representations to capture stable, participant-specific physiological patterns while remaining invariant to short-term noise and temporal perturbations. The method uses participant-level augmentation and momentum-based contrastive training with a regularized InfoNCE loss. A similar idea has been applied to their work, WBM (Erturk et al., 2025), with more modalities. In our settings, we adopt this idea as a unimodal self-supervised baseline applied to PPG signals.

**DINO** (Caron et al., 2021)   DINO is a self-supervised distillation framework that learns representations without labels via a teacher–student paradigm. In our multimodal setting, we instantiate DINO with an ECG-based teacher and a PPG-based student, where the student is trained to match the teacher's output distribution under different data augmentations. Both teacher and student are implemented as 1D Vision Transformers.

**LSM** (Narayanswamy et al., 2024)   LSM proposes a large-scale foundation model for multimodal wearable sensing. The method uses a Vision Transformer backbone trained with masked autoencoding and random masking to learn general-purpose representations. The pretrained model is shown to transfer across a variety of downstream tasks in physiological sensing and human activity recognition. In our experiments, we follow the LSM training protocol and replicate its multimodal design using ECG and PPG modalities only.

**SimCLR** (Chen et al., 2020)   SimCLR establishes contrastive learning as a competitive self-supervised paradigm. The core idea is to maximize agreement between augmented views of the same signal in a latent space while pushing apart representations of different instances. We adapt this paradigm by maximizing agreement between paired ECG and PPG.

**Apple-M** (Fang et al., 2024)   Apple-M proposes a multimodal foundation modeling framework for physiological signals,

**Table 10.** Hyperparameter for pretraining *xMAE* and baselines.

| Configuration | *xMAE* | MAE-1D | MSN | DINO | APPLE | LSM | APPLE-M | PaPaGei-P | SimCLR |
|---|---|---|---|---|---|---|---|---|---|
| Training Epoch | 37 | | | | | | | | |
| Early Stop Patience Epoch | 17 | | | | | | | | |
| Batch Size | 512 | | | | | | | | |
| Warmup Ratio | 10% (of training steps) | | | | | | | | |
| Optimizer | AdamW (Loshchilov & Hutter, 2017) | | | | | | | | |
| Optimizer Momentum $[\beta_1, \beta_2]$ | [0.9,0.95] | | | | | | | | |
| Base Learning Rate | $3e^{-4}$ | | | | | | | | |
| Weight Decay | $1e^{-2}$ | | | | | | | | |
| Learning rate schedule | Linear warmup + cosine scheduler | | | | | | | | |
| Input Modality | PPG+ECG | PPG | PPG | PPG+ECG | PPG | PPG+ECG | PPG+ECG | PPG | PPG+ECG |
| Input Resolution | 1D signal @ 100 Hz $\times$ 10 s | | | | | | | | |
| Random Seed | 77 | | | | | | | | |

using a masked autoencoding objective to pretrain a shared encoder on diverse, synchronized modalities. The pretraining strategy enforces cross-modal reconstruction and includes input modality dropout to encourage integrated representations across signal types. In our work, we adopt Apple-M as a multimodal baseline and apply its pretraining protocol to ECG and PPG signals.

**PaPaGei (Open-Source Weights)** (Pillai et al., 2024)     For this baseline, we directly use the official PaPaGei implementation and publicly released pretrained weights[1], and evaluate the model on our downstream datasets without additional pretraining, enabling a direct comparison with *xMAE*.

**Chronos-Bolt-Tiny (Open-Source Weights)** (Ansari et al., 2024)     For this baseline, we use the official Chronos-Bolt-Tiny implementation and publicly released pretrained weights from Huggingface[2], and directly evaluate the model on our downstream datasets without additional finetuning.

**AnyPPG (Open-Source Weights)** (Nie et al., 2025)     AnyPPG is a CLIP-based foundation model for PPG and ECG on over 100,000 hours of paired PPG–ECG recordings. For this baseline, we adopt the official AnyPPG implementation and released pretrained weights[3]. We use the pretrained PPG encoder as provided, and evaluate its representations on our downstream tasks without additional pretraining or task-specific adaptation.

**PulsePPG (Open-Source Weights)** (Saha et al., 2025)     For this baseline, we adopt the official PulsePPG implementation and released pretrained weights[4], and evaluate the model on our downstream tasks without further pretraining or adaptation.

### D.4. Pretraining Protocols

To ensure a fair comparison across methods, we adopt a largely unified training configuration for *xMAE* and for the baselines that are trained from scratch. Specifically, we align the optimizer, AdamW (Loshchilov & Hutter, 2017), learning rate schedule (linear warmup followed by cosine scheduler), batch size, number of training epochs, and input data resolution across models whenever possible. For baselines that utilize only PPG during pretraining, we compensate for the reduced modality coverage by doubling the effective data volume through augmentation, including random amplitude scaling (in the range [0.8, 1.2]) and signal flipping. For multimodal models, we apply stochastic on-the-fly augmentations during training without changing the training data size. These training details are summarized in Table 10. All training and evaluation are performed on NVIDIA H200 GPUs.

## E. Evaluation Datasets, Tasks and Protocols

In this section, we introduce datasets, tasks, and protocols that are employed for evaluation.

---

[1] https://github.com/Nokia-Bell-Labs/papagei-foundation-model

[2] https://huggingface.co/amazon/chronos-bolt-tiny

[3] https://github.com/Ngk03/AnyPPG

[4] https://github.com/maxxu05/pulseppg

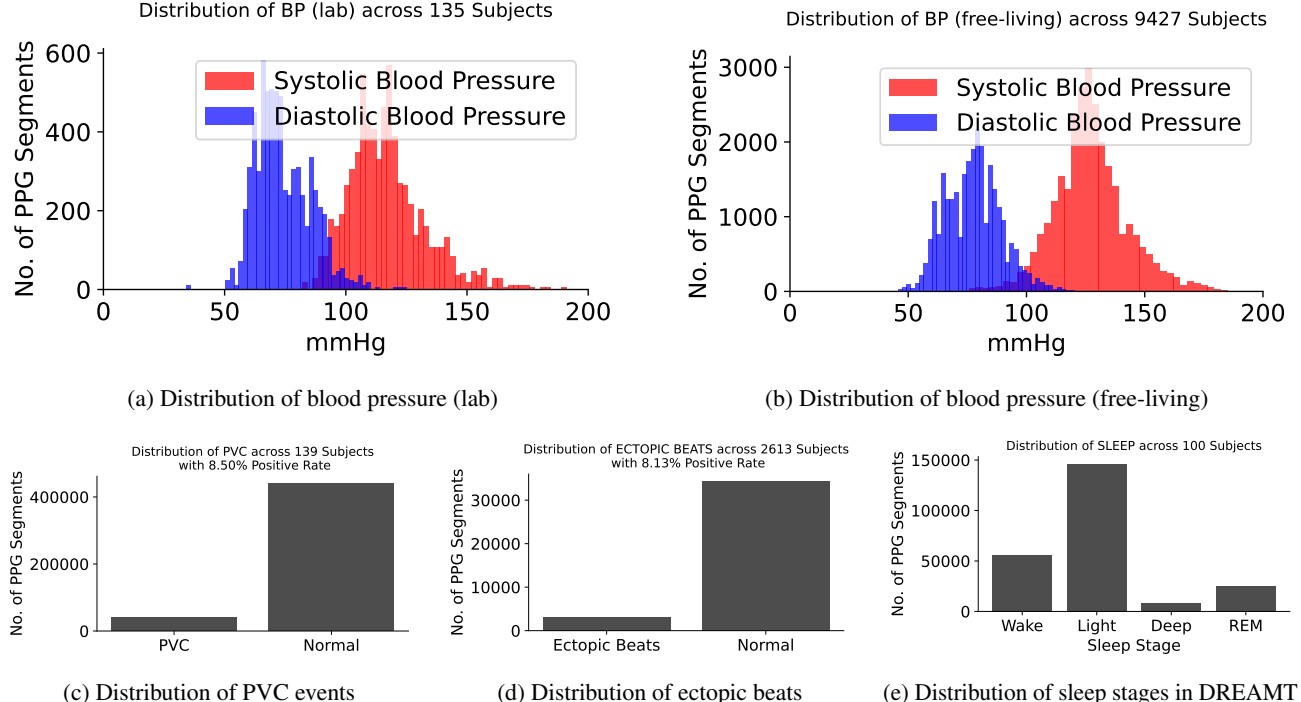

Figure 9. (a)–(b) blood pressure in lab and free-living settings, (c) PVC events, (d) ectopic beats, and (e) sleep stages.

### E.1. Evaluation Datasets and Tasks

In total, we have 19 tasks from 6 datasets, including classification and regression. All datasets analyzed in this project were collected under informed consent, and we will provide relevant information as requested. Notably, these datasets are different from the pretraining dataset in terms of subjects, signal capture devices, and environments. These evaluation protocols follow prior work (Lee et al., 2025).

**Blood Pressure (lab)**    This dataset contains 6966 10-s PPG segments from 135 subjects. We bring in the subjects and collect their PPG readings from smartwatches when the subjects are at the resting stage. The blood pressure measurements are taken from a continuous finger clamp device (CNAP), adjudicated by dual auscultation. We plot the distributions of the blood pressure measurements as shown in Figure 9(a). We utilize this dataset for stage-I hypertension classification, termed as *Hypertension (lab)*, and blood pressure regression for both *Systolic* and *Diastolic* Blood Pressure (BP) with the following details.

- **Hypertension (lab)**: We utilize the standards from stage-I hypertension classification (Cifu & Davis, 2017) where systolic BP $\geq 130$ or diastolic BP $\geq 80$ are classified as hypertension. This results in 1320 segments from 64 subjects.

- **Systolic BP**: Out of 135 subjects, the mean Systolic BP is 116.4 mmHg with a standard deviation of 16.5 mmHg.

- **Diastolic BP**: Out of 135 subjects, the mean Diastolic BP is 74.6 mmHg with a standard deviation of 11.1 mmHg.

**Blood Pressure (free-living)**    This dataset contains 28344 10-s PPG segments from 9427 subjects. The PPG readings are collected from smartwatches during the free-living of the subjects. The blood pressure measurements are reported from cuff-based BP estimation software. We plot the distributions of the blood pressure measurements as shown in Figure 9(b). We utilize this dataset for stage-I hypertension classification, termed as *Hypertension (free-living)*, and blood pressure regression for both *Systolic* and *Diastolic* Blood Pressure (BP) with the following details.

- **Hypertension (free-living)**: Stage-I hypertension has 15412 segments from 5979 subjects.

- **Systolic BP**: Out of 9427 subjects, the mean Systolic BP is 127.2 mmHg with a standard deviation of 15.7 mmHg.

- **Diastolic BP**: Out of 9427 subjects, the mean Diastolic BP is 77.6 mmHg with a standard deviation of 11.9 mmHg.

**AFib**     This dataset is collected in a lab setting where an ECG patch (manufactured by Preventice Solutions, Inc.) is attached to subjects for ground truth labeling. In the meantime, PPG signals are collected from smartwatches. As shown in Figure 9(c), there are 480717 10-s PPG segments from 139 subjects. We utilize this dataset for *Premature Ventricular Contractions (PVCs)* detection with the following details.

- **PVC**: PVCs are abnormal beats arising in the ventricles (Cha et al., 2012; Kaya & Pehlivan, 2015). We use paired PPG–ECG data, with ECG annotations generated using BeatLogic (Teplitzky et al., 2020) and manually verified. This task evaluates whether ubiquitous PPG can approximate arrhythmia detection typically restricted to ECG. Out of 480717 segments, 8.5% segments are labeled as PVCs.

**MX**     This dataset is collected in a free-living setting where users are wearing smartwatches for PPG collection. As shown in Figure 9(d), there are 37309 segments from 2613 subjects. We utilize this dataset for *Ectopic Beats* detection with the following details.

- **Ectopic Beats**: Ectopic beats are extra or skipped heartbeats caused by a brief misfire in the heart's electrical system, making people feel a flutter, thump, or skipped beat. They're common, usually harmless, and often triggered by stress, caffeine, alcohol, lack of sleep, or electrolyte imbalance. While often benign, frequent ectopic beats or beats accompanied by dizziness, chest pain, or fainting signal underlying health issues. Out of 37309 segments, 8.13% segments are labeled as ectopic beats.

**DREAMT**     DREAMT (Wang et al., 2025) is a sleep staging dataset hosted on PhysioNet, which includes overnight wristband data with simultaneous polysomnography (PSG) and PPG. In total, there are 235419 10-s PPG segments from 100 subjects. Annotations follow American Academy of Sleep Medicine (AASM) standards into wake, Rapid eye movement (REM), Non-rapid eye-movement (NREM) stage 1 (NREM1), NREM2, and NREM3, excluding missing and preparation segments. Following the standards (Berry et al., 2020), we combine NREM1 and NREM2 as *light* stage, and refer to NREM3 as *deep* stage. We provide the breakdown in Figure 9(e). We note that sleep staging has canonically been designed by leveraging the whole sleep cycle, but we are assessing the ability to monitor real-time sleep staging from much shorter PPG segments (10 seconds). We utilize this dataset to examine whether PPG encodes temporal patterns sufficient for *Sleep Stage* binary classification with the following details.

- **Wake**: This is the *Wake* stage with 56127 segments (23.8%).

- **Light**: This is the stage of *Light*, combining labels of NREM1 and NREM2. There are 146085 segments (62.1%).

- **Deep**: This is the stage of *Deep* from the label of NREM3. In total, there are 8112 segments (3.4%).

- **REM**: This is the *REM* stage with 25095 segments (10.7%).

**Abnormal Lab Test**     For abnormal lab test prediction (Figure 10), we use Watch PPG collected at REDACTED University paired with clinical laboratory results. Each test is framed as a binary classification task. PPG preprocessing matches other tasks. Targets include *A1C*, *hemoglobin*, *platelets*, and *sodium*, each selected for established clinical relevance. This task extends evaluation beyond cardiovascular and behavioral endpoints to systemic markers of metabolic, renal, and hematologic health. We note that it is unclear whether PPG can predict abnormal from healthy lab values based on the PPG alone. Despite this, the university presents us with an opportunity to discover if PPG signals can indicate these lab tests, making this an exploratory task in our benchmark. We provide the detailed description for these lab tests as follows:

- **A1C (Glycated Hemoglobin)**: A1C measures average blood glucose levels over the past 2–3 months. It is the primary diagnostic tool for diabetes and a key indicator for managing long-term blood sugar control. Elevated A1C levels are linked to increased risk of cardiovascular disease, kidney damage, and other complications. Out of 5242 10-s PPG segments from 19 subjects, 52.79% are labeled as positive.

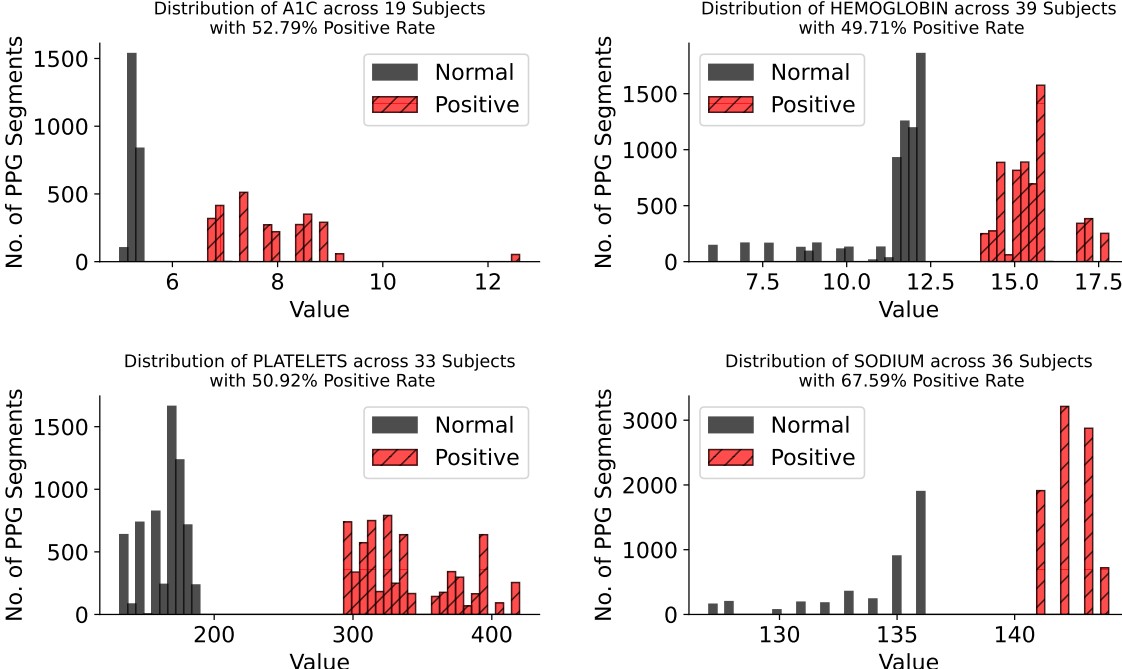

**Figure 10.** Distribution of Abnormal Lab Tests.

- **Hemoglobin**: Hemoglobin indicates an oxygen-carrying protein in red blood cells. Low levels indicate anemia, while elevated levels may suggest polycythemia vera. Out of 12933 10-s PPG segments from 39 subjects, 49.71% are labeled as positive.

- **Platelets**: Platelets is critical for clotting. While low count (thrombocytopenia) increases bleeding risk, high count (thrombocytosis) increases clot risk. Out of 12975 10-s PPG segments from 33 subjects, 50.92% are labeled as positive.

- **Sodium**: Sodium regulates fluid balance and blood pressure. Abnormalities can indicate dehydration, renal disease, or endocrine disorders. Out of 12903 10-s PPG segments from 36 subjects, 67.59% are labeled as positive.

**Demographics** We also evaluate the quality in terms of demographics data, such as *Age* and *Body Mass Index (BMI)* based on the *Blood Pressure (lab)* and *Blood Pressure (free-living)* datasets. Note that we only keep the PPG segments where the subjects' demographics are available.

- **Age (lab)**: As shown in Figure 11(a), there are 9263 segments from 63 subjects. The mean age is 32.2 with a standard deviation of 9.8.

- **Age (free-living)**: As shown in Figure 11(b), there are 27697 segments from 9149 subjects. The mean age is 44.4 with a standard deviation of 12.7.

- **BMI (lab)**: As shown in Figure 11(c), there are 9263 segments from 63 subjects. The mean BMI is 25.0 with a standard deviation of 4.6.

### E.2. Evaluation Protocols

To evaluate the quality of learned PPG representations independent of end-to-end finetuning, we adopt a *linear probing* protocol, following (Pillai et al., 2024). In this setting, the backbone PPG encoder is frozen, and task-specific predictors are trained solely on top of the extracted embeddings. This protocol isolates the expressiveness and task relevance of the learned representation. For each task, we employ 5-fold cross-validation based on subjects. In each iteration, 80% subjects are used

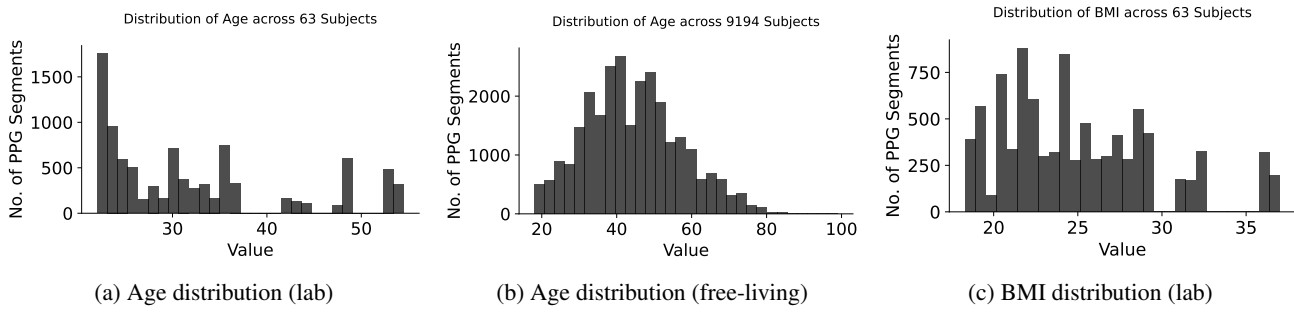

(a) Age distribution (lab)   (b) Age distribution (free-living)   (c) BMI distribution (lab)

**Figure 11.** Distribution of demographics.

for training and validation, and the rest 20% subjects are used for testing. Performance is evaluated on the held-out subjects and is reported as the mean and standard deviation across all five folds.

**Embedding Extraction**  Given an input PPG segment, we pass it through the pretrained PPG encoder and extract the embedding from the final representation layer. The encoder weights remain frozen throughout all downstream experiments, unless stated otherwise.

**Classification Linear Probing**  For classification tasks, we train a simple classifier on top of the frozen embeddings, following common linear probing practice (Pillai et al., 2024). The classifier is trained using only the training split embeddings and evaluated on held-out test subjects' embeddings without any encoder updates.

**Regression Linear Probing**  For regression tasks, we follow an analogous protocol as defined in classification tasks. Performance is reported using a standard regression metric (i.e., mean absolute error), computed on the held-out test set across *xMAE* and all baselines.

**Few-Shot Finetuning**  In addition to linear probing, we evaluate representation adaptability under limited supervision using a *k-shot finetuning* protocol. This setting simulates realistic low-data scenarios commonly encountered in personalized and clinical applications. For the task of PVC detection, we attach a lightweight task-specific classifier consisting of a 2-layer multilayer perceptron (MLP) on top of the pretrained encoder. Unlike linear probing, all parameters are updated during training. To construct the k-shot training set, we randomly sample $k$ labeled PPG segments *per subject* in the training split. Importantly, the selected samples vary across different values of $k$, ensuring that each k-shot setting reflects a realistic change in data availability rather than the reuse of the same segments. The test set is held fixed across all k-shot experiments and remains identical to that used in linear probing. This design ensures that performance differences across different values of $k$ are attributable solely to changes in the amount of labeled training data, rather than variations in evaluation data. The results are reported in Figure 6. We kept the hyperparameters, such as learning rate (1e-5), batch size (2048) same across models.

**Random Seed**  We set the random seed to 1 across all tasks and evaluations.

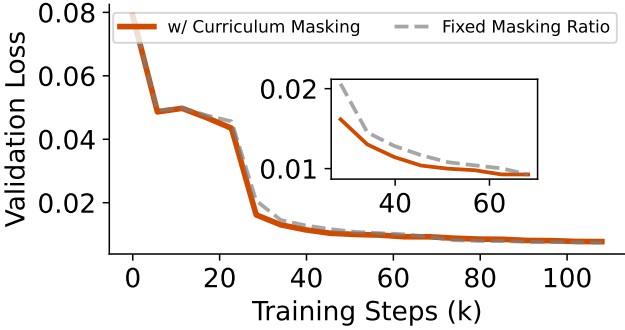

**Figure 12.** Validation loss under curriculum masking and fixed high masking ratio ($M$=90%). Curriculum masking yields smoother and faster early convergence while achieving comparable final loss.

# F. Justification of Curriculum ECG Masking

We provide a justification of our choice on curriculum ECG masking. Let $\mathcal{L}(M;\theta)$ denote the masked cross-modal reconstruction loss under ECG masking ratio $M \in (0,1)$ and model parameters $\theta$. In masked reconstruction, increasing $M$ strictly reduces the amount of visible ECG context, thereby increasing task difficulty:

$$\mathcal{L}(M_1;\theta) \leq \mathcal{L}(M_2;\theta), \quad \text{for } M_1 < M_2 \text{ and fixed } \theta, \tag{13}$$

since fewer observations are available to reconstruct the masked signal, and biosignals such as ECG exhibit strong temporal self-correlation (Yu et al., 2006).

We adopt a curriculum learning strategy over the ECG masking ratio to progressively encourage cross-modal reasoning. Concretely, we initialize training with a lower masking ratio $M_0 = 80\%$ and increase the masking ratio in fixed increments of $\Delta M = 5\%$ when the relative improvement in reconstruction loss exceeds a predefined threshold (i.e., 10%):

$$M \leftarrow M + \Delta M \quad \text{if} \quad \frac{\mathcal{L}_{\text{prev}} - \mathcal{L}_{\text{curr}}}{\mathcal{L}_{\text{prev}}} \geq 0.10, \tag{14}$$

until reaching a maximum masking ratio of $M_{\max} = 90\%$, which preserves at least one full cardiac cycle. Following prior work in vision (He et al., 2022), we initialize the curriculum at a high masking ratio (80%) because lower masking regimes allow a substantial shortcut based on ECG-only completion from visible context.

With contiguous masking, when a large fraction of the ECG remains visible, the masked segment can often be reconstructed via within-signal inpainting or extrapolation from adjacent ECG context, without requiring meaningful use of PPG. Starting at 80% masking substantially limits this ECG-only shortcut while preserving enough visible context for stable optimization. We then progressively increase the masking ratio to 90% in 5% steps to further suppress within-signal completion and shift the learning objective toward cross-signal inference grounded in the ECG–PPG relationship. The purpose of this curriculum is not to finely tune the exact masking schedule, but to enforce a controlled transition from ECG-only completion to cross-modal reasoning, which we find sufficient to induce stable training and strong downstream performance.

From an optimization perspective, this curriculum can be viewed as a continuation method, where the model is trained on a sequence of increasingly difficult objectives $\{\mathcal{L}(M_t;\theta)\}$. Early stages benefit from better-conditioned gradients due to greater ECG visibility, while later stages progressively reduce ECG context and increase reliance on PPG signals. From a modeling perspective, decreasing ECG visibility shifts reconstruction from morphology-driven interpolation toward physiologically grounded cross-signal inference, encouraging the model to encode ECG–PPG transition relationships.

Figure 12 compares the validation loss trajectories of our curriculum masking and a fixed high masking ratio ($M$=90%) during pretraining. While both approaches converge to comparable final loss values, their optimization behaviors differ during training. The curriculum-based model exhibits a faster and more stable loss decrease, particularly in early and mid-training, whereas the fixed-mask model shows slower progress under the more difficult objective. As the masking ratio is gradually increased, the curriculum model continues to reduce loss without abrupt degradation, indicating successful adaptation to the increasing task difficulty. These results suggest that curriculum masking primarily improves training stability and convergence behavior. We believe this strategy provides a principled mechanism to align the learning process with the intended cross-modal reasoning objective.

Table 11 demonsrates curriculum-masked pretraining tends to make downstream performance more stable across data splits by reducing sensitivity to any single split. In our results, this shows up most clearly on *Hypertension (lab)*, where the standard deviation across splits drops from 7.3 to 4.8 when switching from a fixed mask ratio to curriculum masking, indicating less split-to-split fluctuation. On *Ectopic Beats*, performance is already stable under fixed masking, and curriculum masking preserves this stability while delivering consistent improvements across all five splits. Together, these patterns suggest that gradually increasing the ECG masking ratio improves robustness by either reducing variance when the task is noisy or maintaining low variance while improving accuracy when the task is already well-behaved.

**Table 11.** Detailed linear-probing performance breakdown (AUROC) under settings of fixed mask ratio and curriculum masking.

| Task | Setup | Split 1 | Split 2 | Split 3 | Split 4 | Split 5 | Average | Std |
|------|-------|---------|---------|---------|---------|---------|---------|-----|
| *Hypertension (lab)* | Fixed Mask Ratio | 66.7 | 76.3 | 54.3 | 66.3 | 71.3 | 66.9 | 7.3 |
| | w/ Curriculum Masking | 64.1 | 76.7 | 64.8 | 66.7 | 72.0 | 68.8 | 4.8 |
| *Ectopic Beats* | Fixed Mask Ratio | 82.1 | 89.3 | 87.3 | 84.8 | 85.4 | 83.4 | 2.4 |
| | w/ Curriculum Masking | 84.1 | 89.9 | 90.5 | 87.6 | 86.8 | 87.8 | 2.3 |

# G. Proof and Evidence on the Effectiveness of Masked Cross-Modal Reconstruction in *xMAE*

## G.1. Why Masked Cross-Modal Reconstruction Encourages Temporal Asymmetry

We model the cardiovascular system using a latent physiological process $\{S_t\}_{t\in\mathbb{Z}}$ that generates ECG and PPG as

$$E_t = g(S_t) + \epsilon_t^E, \tag{15}$$

$$P_t = h(S_{t-\Delta}) + \epsilon_t^P, \tag{16}$$

where $\Delta \in \mathbb{N}$ is the physiological ECG–PPG delay (e.g., pulse arrival time), $\epsilon_t^E, \epsilon_t^P$ are independent zero-mean noise, and $g(\cdot)$ and $h(\cdot)$ are measurement functions (e.g., electrical sensing). ECG reflects the instantaneous electrical activation $S_t$, while PPG reflects a delayed mechanical response. This inherent asymmetry is key to cardiovascular dynamics (Block et al., 2020).

**Multimodal MAE objective.** Let $\mathcal{V}^E, \mathcal{M}^E$ denote the visible and masked ECG indices, and $\mathcal{V}^P, \mathcal{M}^P$ the corresponding PPG sets. A standard multimodal masked autoencoder (MM-MAE) encodes the visible tokens,

$$H = \text{Enc}_\theta(E_{\mathcal{V}^E}, P_{\mathcal{V}^P}),$$

where $H$ is the latent representation and reconstructs the masked ones,

$$\hat{E}_t = \text{Dec}_\theta^E(H), \qquad \hat{P}_t = \text{Dec}_\theta^P(H),$$

by minimizing

$$\mathcal{L}_{\text{MM-MAE}} = \mathbb{E}\left[\sum_{t\in\mathcal{M}^E} \|\hat{E}_t - E_t\|^2 + \sum_{t\in\mathcal{M}^P} \|\hat{P}_t - P_t\|^2\right]. \tag{17}$$

Under unlimited model capacity, the Bayes–optimal reconstructions are

$$\phi^E(E_{\mathcal{V}^E}, P_{\mathcal{V}^P}) = \mathbb{E}[E_{\mathcal{M}^E} \mid E_{\mathcal{V}^E}, P_{\mathcal{V}^P}], \tag{5}$$

$$\phi^P(E_{\mathcal{V}^E}, P_{\mathcal{V}^P}) = \mathbb{E}[P_{\mathcal{M}^P} \mid E_{\mathcal{V}^E}, P_{\mathcal{V}^P}], \tag{6}$$

since conditional expectation uniquely minimizes squared error.

**Multimodal MAE does not require modeling the delay.** To understand whether MM-MAE must learn the physiological delay $\Delta$, we introduce a mild assumption reflecting a common empirical property of biosignals: ECG and PPG are each highly predictable from their own nearby samples (Yu et al., 2006).

**Assumption G.1** (Local self-sufficiency of each modality)**.** There exist neighborhoods $\mathcal{N}^E(t)$ and $\mathcal{N}^P(s)$ such that, for all masked positions $t \in \mathcal{M}^E$ and $s \in \mathcal{M}^P$,

$$E_t \perp P_{\mathcal{V}^P} \mid E_{\mathcal{N}^E(t)}, \tag{18}$$

$$P_s \perp E_{\mathcal{V}^E} \mid P_{\mathcal{N}^P(s)}. \tag{19}$$

That is, once a small local window of ECG around $t$ is known, PPG provides no additional information about $E_t$; and symmetrically for PPG.

Assumption G.1 reflects the strong morphological and temporal regularity of ECG and PPG (e.g., predictable QRS or systolic peaks). In typical MAE masking schemes, each masked token usually retains some visible neighbors, making this assumption practical. Assumption G.1 is not intended to be physiologically exact, but rather a sufficient condition illustrating that MM-MAE admits optimal solutions that ignore cross-modal timing whenever local self-predictability dominates.

Then, we obtain

$$E_{\mathcal{M}^E} \perp P_{\mathcal{V}^P} \mid E_{\mathcal{V}^E}, \qquad P_{\mathcal{M}^P} \perp E_{\mathcal{V}^E} \mid P_{\mathcal{V}^P}. \tag{20}$$

Applying the tower property of conditional expectation to (5)–(6) with (20) yields

$$\phi^E(E_{\mathcal{V}^E}, P_{\mathcal{V}^P}) = \mathbb{E}[E_{\mathcal{M}^E} \mid E_{\mathcal{V}^E}], \tag{8}$$

$$\phi^P(E_{\mathcal{V}^E}, P_{\mathcal{V}^P}) = \mathbb{E}[P_{\mathcal{M}^P} \mid P_{\mathcal{V}^P}]. \tag{9}$$

Equations (8)–(9) show that optimal MM-MAE solutions exist in which **each modality reconstructs itself solely from its own visible tokens**. Therefore, it admits solutions that ignore cross-signal timing, including the ECG–PPG delay $\Delta$, yet still achieve the global minimum of (17).

**Proposition G.2** (Multimodal MAE does not require modeling the delay). *Under Assumption G.1, the MM-MAE objective* (17) *admits global minimizers of the form (8)–(9), in which ECG reconstructs ECG and PPG reconstructs PPG without using cross-modal information. These minimizers achieve identical risk for all $\Delta$, implying that the ECG–PPG delay are not required under the MM-MAE objective.*

This formalizes the empirical phenomenon: multimodal MAEs tend to rely on within-modality structure and have no inherent incentive to learn the delayed physiological coupling between ECG and PPG.

**Masked Cross-Modal reconstruction MAE (*xMAE*).**    In contrast, *xMAE* reconstructs masked ECG *from PPG and only limited ECG context*:

$$\hat{E}_{\mathcal{M}^E} = f_\theta(P_{1:T}, E_{\mathcal{V}^E}),$$

by minimizing

$$\mathcal{L}_{\mathrm{xMAE}} = \mathbb{E}\left[\sum_{t \in \mathcal{M}^E} \|\hat{E}_t - E_t\|^2\right].$$

The Bayes–optimal predictor is

$$f_\Delta^*(P_{1:T}, E_{\mathcal{V}^E}) = \mathbb{E}_\Delta[E_{\mathcal{M}^E} \mid P_{1:T}, E_{\mathcal{V}^E}],$$

which depends nontrivially on the true delay $\Delta$, because PPG at time $t$ informs the latent state $S_{t-\Delta}$ that determines $E_t = g(S_t)$. Changing $\Delta$ changes this conditional expectation. If the model uses an incorrect delay $\Delta' \neq \Delta$, the reconstructed R-peaks will be systematically misaligned in expectation, leading to increased reconstruction error.

**Proposition G.3** (Identifiability of delay under cross-modal reconstruction). *Under this model, the Bayes–optimal cross-modal reconstruction predictor depends nontrivially on the physiological delay $\Delta$. Any predictor that fails to encode the correct delay incurs higher expected reconstruction risk. Thus, $\Delta$ is identifiable under the* xMAE *objective.*

**Implications.**    Multimodal MAE is symmetric and permits solutions that rely primarily on within-modality correlation. In contrast, *xMAE* is directional and physiologically grounded: reconstructing ECG from PPG forces the model to encode the electrical-to-mechanical relationships and the associated ECG–PPG delay. To our knowledge, this form of asymmetric cross-modal reconstruction has not been explored as an inductive bias for biosignal representation learning.

### G.2. An Inductive Bias Perspective on Cross-Modal Reconstruction in *xMAE*

We provide an inductive-bias interpretation of *xMAE* to clarify how its training objective differs from standard multimodal representation learning. This perspective makes explicit what structural assumptions are encouraged by cross-modal reconstruction and why they are well matched to the ECG–PPG relationship.

ECG and PPG arise from a shared underlying cardiac process but are observed through different transformations. ECG reflects electrical activation, while PPG is a delayed and transformed hemodynamic response shaped by vascular transport

and peripheral sensing. Importantly, this mapping is neither temporally aligned nor information preserving, as multiple electrical states may induce similar peripheral waveforms, and the delay varies across individuals and physiological states.

Most multimodal representation learning methods implicitly assume conditional exchangeability between modalities and therefore rely on objectives such as joint reconstruction or contrastive alignment. These formulations are effective when modalities provide approximately co-temporal views of the same latent state, but they impose an inductive bias that is misaligned with the ECG–PPG relationship, which is temporally directional.

*xMAE* introduces a different inductive bias by formulating pretraining as inference under partial observability of the upstream signal. Given full access to the downstream PPG signal and a limited visible subset of ECG, the model is trained to reconstruct masked ECG segments. This objective biases the learned PPG representations toward capturing stable and informative aspects of the electrical-to-mechanical relationships, such as relative timing (Block et al., 2020), rather than modality-specific self-correlation.

Curriculum masking (Appendix F) further sharpens this inductive bias. At low masking ratios, reconstruction is supported by local ECG context, stabilizing optimization. As masking increases, successful reconstruction increasingly requires exploiting delayed temporal and morphological cues present in PPG, preventing trivial reconstruction from ECG alone.

From this perspective, *xMAE* encourages PPG representations to encode low-dimensional, physiologically meaningful functionals of the cardiac activity that are preserved through vascular transport, such as beat timing, inter-beat intervals, and pulse arrival dynamics. These quantities are central to downstream cardiovascular tasks and are precisely the aspects of physiology that prior pretraining objectives tend to underemphasize.

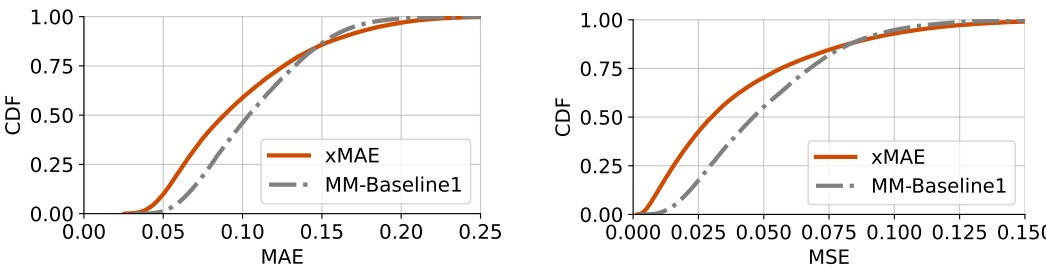

**Figure 13.** ECG reconstruction errors between *xMAE* and MM-baseline. (Left) Mean Absolute Error. (Right) Mean Squared Error.

### G.3. Evidence 1: ECG Reconstruction Error between *xMAE* and Multimodal MAE Baseline

We pretrain *xMAE* and a multimodal MAE baseline (we term it MM-Baseline1) with $\approx$ 3.4M 10-s paired ECG and PPG segments from 2.4k users. We held out a different set of users for the reconstruction task. Next, we explain the details in terms of the pretraining masking strategy and objectives.

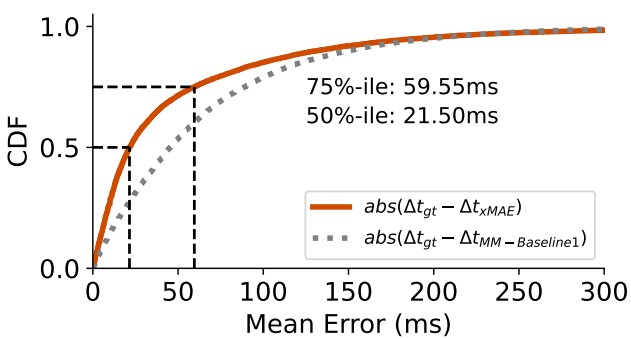

**Figure 14.** Cumulative distribution of absolute time-delay error between the ground-truth ECG–PPG delay ($\Delta t_{gt}$) and delays estimated from reconstructed signals. We quantify this delay by the time difference between the ECG R-peak and the PPG onset valley. *xMAE* exhibits consistently lower delay error and tighter alignment than the baseline across 31k 10-s segments.

**Setup for *xMAE*** For a pair of PPG and ECG, we mask out ECG with continuous temporal masks covering 90% of ECG as detailed in §3, and we do not mask out PPG. The objective is to reconstruct the ECG on its masked portion.

**Setup for MM-Baseline1** To build this baseline, we follow how prior works (Narayanswamy et al., 2024; Fang et al., 2024) pretrain with multiple modalities as follows. For a pair of PPG and ECG, we randomly mask out 90% of ECG and 60% of PPG. The objective is to reconstruct the ECG and PPG on their masked portions.

In inference, we follow the same masking strategy for ECG as defined during pretraining and leave PPG unmasked, and the reconstructed ECG is used to calculate error with ground truth ECG.

**Analysis** Figure 13 depicts the errors of *xMAE* and baselines. We observe that *xMAE* consistently achieves lower reconstruction error compared to other baselines across 31k segments from held-out users. This improvement is particularly notable given that *xMAE* solves a more challenging reconstruction task: ECG is masked with long continuous temporal blocks and must be reconstructed without access to any local ECG context. In contrast, MM-Baseline1 applies random masking, which allows the model to exploit nearby unmasked ECG samples through local interpolation. The lower error of *xMAE* therefore indicates that it learns a more informative cross-modal structure between ECG and PPG, rather than relying on intra-modal shortcuts. Overall, these results suggest that the masked cross-modal reconstruction objective in *xMAE* more effectively captures the temporal relationship between modalities, enabling more accurate reconstruction of ECG from PPG and a limited context of ECG. We further provide a number of qualitative results on these models for ECG reconstruction as shown in Figure 15 and Figure 16.

### G.4. Evidence 2: *xMAE* Captures the Time Delay Better than Multimodal Baselines

Figure 14 evaluates how well different models preserve the physiological time delay between ECG and PPG by comparing the absolute error between the ground-truth delay, $\Delta t_{gt}$, computed from real ECG–PPG pairs, and the delay estimated from reconstructed signals. Using Neurokit2 (Makowski et al., 2021), we quantify this delay as the time difference between the ECG R-peak and the PPG onset valley, which serves as a meaningful proxy for ECG–PPG temporal delay. As shown in the figure, *xMAE* consistently yields smaller delay errors than both baselines across the held-out users (31k segments). In particular, the *xMAE* curve rises more steeply near zero error, indicating that a larger fraction of samples exhibit small delay deviations from the ground truth. In contrast, both baselines show heavier tails, suggesting higher variance and less stable temporal alignment. The median error is 21.5 ms and 45.5 ms for *xMAE* and MM-Baseline1, respectively, suggesting 53.3% improvement. These results demonstrate that *xMAE* more accurately captures the cross-modal structure between ECG and PPG, supporting the claim that its cross-reconstruction objective encourages learning of physiological temporal relationships rather than relying on intra-modal shortcuts (i.e., interpolation based on neighboring signals).

### G.5. Distinction from PPG-to-ECG Generation

Recent work has explored generating ECG waveforms from PPG signals for synthesis or augmentation (Sarkar & Etemad, 2021; Kong et al., 2024; Fang et al., 2025). These methods are designed to optimize waveform realism, treating ECG generation as the end goal and evaluating success through reconstruction fidelity.

*xMAE* takes a fundamentally different perspective. We use ECG as a training-time supervisory signal to shape PPG representations. Our masked cross-modal reconstruction objective enforces a structural inductive bias: it requires the model to reason over the temporal and directional relationship between modalities, instead of optimizing for signal-level reconstruction quality.

As a result, *xMAE* learns PPG representations that transfer robustly across tasks, datasets, and sensing conditions, even when ECG is entirely absent at deployment. These findings position *xMAE* not as a PPG-to-ECG generation model, but as a representation learning framework for multimodal settings where signals observe different, temporally ordered stages of a shared process.

## H. Extended Analysis

### H.1. Additional Results Against Open-Source Models

We present the performance of all tasks in this section in Table 4, Table 12, and Table 13. Again, these models are trained with different architectures, different sizes, and different pretraining datasets. Yet, *xMAE* consistently achieves comparable

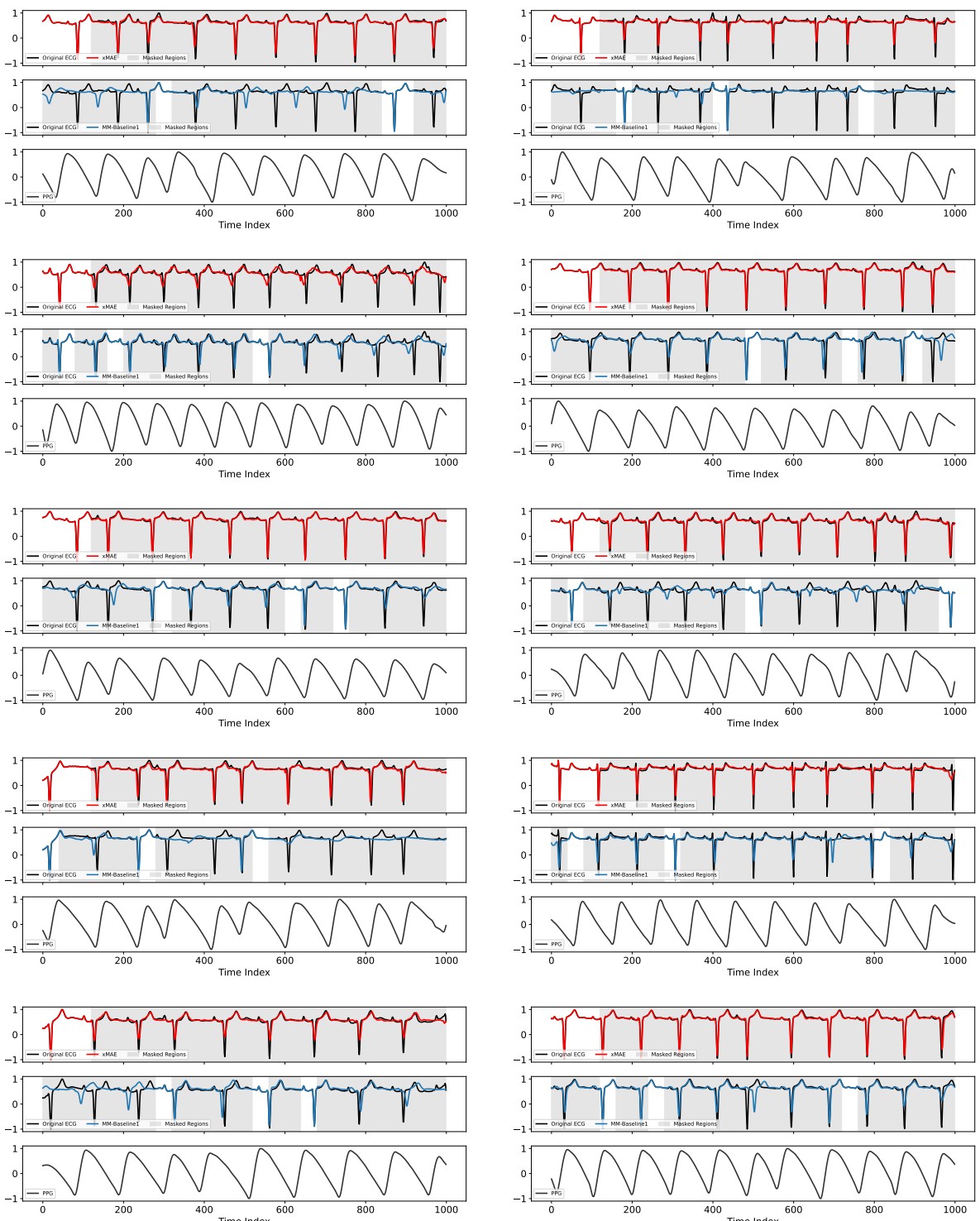

**Figure 15.** ECG reconstruction illustrations.

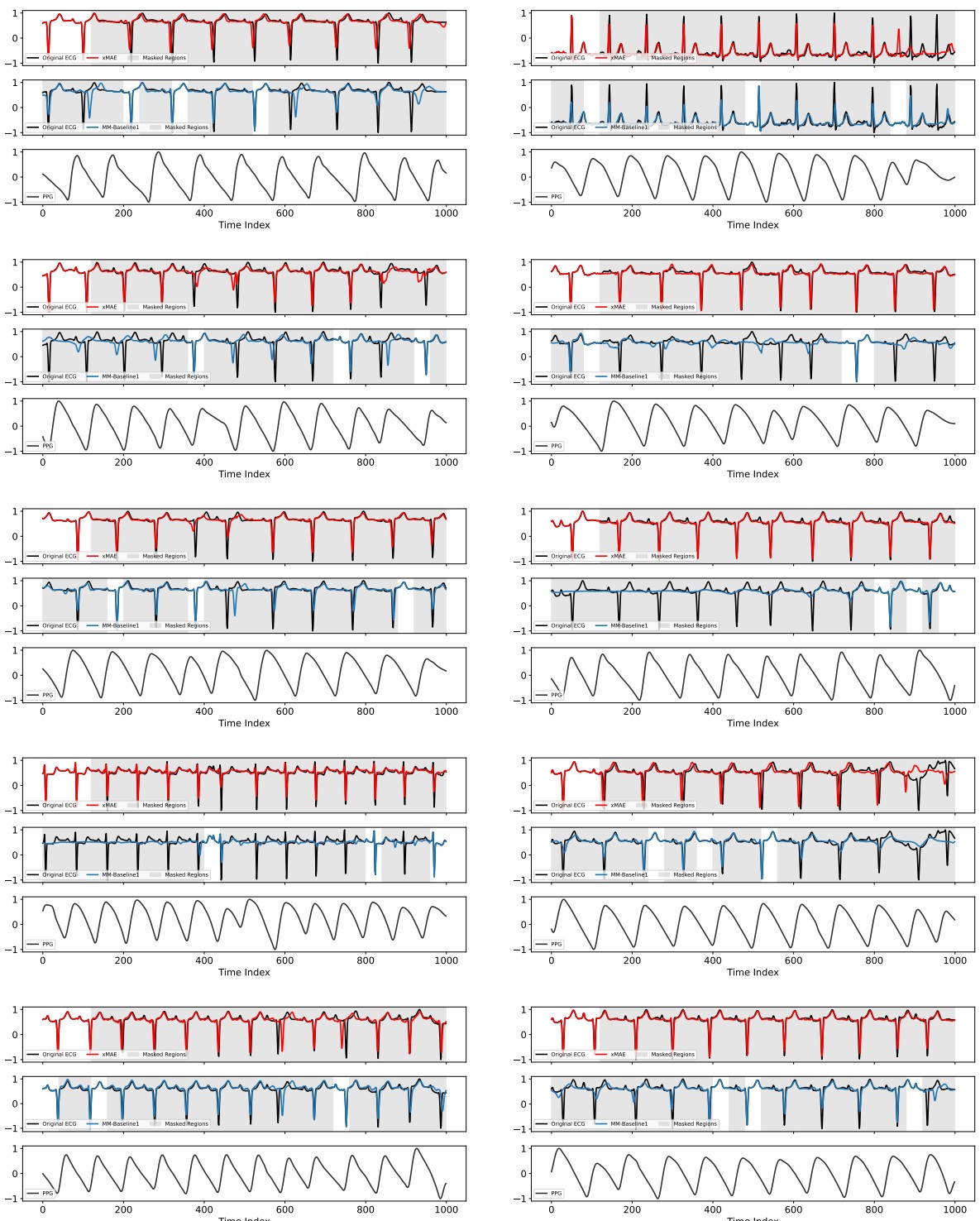

**Figure 16.** ECG reconstruction illustrations.

performance on clinically and physiologically grounded tasks, particularly cardiovascular outcomes and laboratory test prediction, where accurate modeling of beat-level timing and pulse dynamics is critical. This demonstrates that pretraining with the right inductive bias leads to efficient biosignal learning.

**Table 12.** Performance comparison against open-source pretrained models on classification (AUROC) tasks using linear probing. *xMAE* achieves competitive or superior performance despite using fewer parameters and less pretraining data.

| Model | Classification (AUROC↑) | | | | | | | |
|---|---|---|---|---|---|---|---|---|
| | Hypertension (lab) | PVC | Hemoglobin | Platelets | Sodium | Light | Deep | REM |
| PaPaGei (Pillai et al., 2024) | 55.9 (±4.5) | 78.8 (±2.0) | 52.2 (±9.7) | 62.5 (±16.8) | 62.3 (±12.7) | 56.8 (±1.1) | 56.2 (±4.2) | 52.7 (±1.8) |
| AnyPPG (Nie et al., 2025) | 65.0 (±5.8) | **82.8** (±4.1) | 49.7 (±9.0) | 67.9 (±17.2) | 63.3 (±17.9) | 56.5 (±1.6) | 56.3 (±3.9) | 52.5 (±1.4) |
| Chronos-Bolt (Ansari et al., 2024) | 56.2 (±6.6) | 81.7 (±4.1) | 51.9 (±12.5) | 67.8 (±19.0) | 60.0 (±19.6) | 57.1 (±1.5) | **56.5** (±3.9) | 51.2 (±2.3) |
| Pulse-PPG (Saha et al., 2025) | 61.7 (±6.7) | 78.9 (±4.5) | 52.6 (±10.2) | 65.7 (±11.6) | 60.1 (±17.9) | 57.3 (±1.8) | 51.9 (±6.0) | 52.1 (±1.2) |
| *xMAE* | **68.8** (±4.8) | 81.4 (±5.1) | **62.0** (±16.0) | **68.6** (±16.5) | 61.7 (±16.1) | **57.5** (±1.3) | 55.9 (±5.3) | **54.5** (±2.5) |

**Table 13.** Performance comparison against open-source pretrained models on regression (MAE) tasks using linear probing. *xMAE* achieves competitive or superior performance despite using fewer parameters and less pretraining data.

| Model | Regression (MAE↓) | | |
|---|---|---|---|
| | Systolic BP (lab) | Diastolic BP (lab) | Age (free-living) |
| PaPaGei (Pillai et al., 2024) | 12.75 (±1.37) | 9.17 (±0.99) | 9.79 (±0.22) |
| AnyPPG (Nie et al., 2025) | 12.46 (±1.63) | 8.86 (±0.88) | 8.92 (±0.21) |
| Chronos-Bolt (Ansari et al., 2024) | 12.63 (±1.45) | 9.07 (±0.99) | 9.05 (±0.19) |
| Pulse-PPG (Saha et al., 2025) | 12.98 (±1.56) | 9.26 (±0.84) | **8.54** (±0.22) |
| *xMAE* | **11.92** (±1.42) | **8.65** (±0.73) | 8.66 (±0.20) |

## H.2. Matched-Domain Pretraining Analysis

To examine whether the fingertip-to-wrist domain shift affects downstream wrist PPG performance, we conduct a matched-domain comparison on wrist PPG tasks. Specifically, we compare two *xMAE* variants:

- **xMAE (Finger):** pretrained using paired ECG and fingertip PPG.

- **xMAE (Wrist):** pretrained using paired ECG and wrist PPG.

We control the pretraining data scale across the two settings to isolate the effect of PPG sensing location rather than dataset size. Table 14 reports the downstream AUROC scores on wrist PPG tasks.

**Table 14.** Matched-domain comparison between *xMAE* pretrained with fingertip PPG and wrist PPG. Results are reported as AUROC, where higher is better.

| Task | *xMAE* (Finger) | *xMAE* (Wrist) |
|---|---|---|
| Hypertension (lab) | 68.8 (±4.8) | 65.9 (±8.6) |
| Hypertension (free-living) | 58.5 (±1.1) | 56.7 (±0.9) |
| Ectopic Beats | 87.8 (±2.3) | 87.4 (±2.7) |
| PVC | 81.4 (±5.1) | 80.5 (±3.6) |
| Wake | 66.4 (±2.3) | 65.9 (±2.4) |
| Light | 57.5 (±1.3) | 57.4 (±1.6) |
| Deep | 55.9 (±5.3) | 54.5 (±6.2) |
| REM | 54.5 (±2.5) | 54.4 (±2.5) |

Overall, *xMAE* achieves similar performance across the two pretraining domains. Although pulse arrival time and signal morphology can differ between fingertip and wrist PPG, the downstream results remain close across all tasks. This suggests

**Table 15.** Data efficiency analysis under different pretraining volumes. Results are reported as AUROC, where higher is better.

| Volume | Model | Hyptn (lab) | Hyptn (free-living) | A1C | Wake | PVC |
|---|---|---|---|---|---|---|
| 3% | *xMAE* | 60.5 (±4.4) | 55.9 (±1.2) | 61.9 (±14.1) | 63.6 (±1.3) | 75.0 (±4.9) |
| | Apple-M | 52.9 (±3.8) | 55.4 (±1.0) | 47.3 (±7.8) | 64.5 (±1.2) | 66.1 (±4.3) |
| | LSM | 51.7 (±7.2) | 54.8 (±0.7) | 45.9 (±9.0) | 64.9 (±1.3) | 69.0 (±3.3) |
| 30% | *xMAE* | 65.6 (±2.7) | 57.4 (±0.7) | 64.6 (±13.2) | 64.9 (±1.1) | 79.8 (±3.8) |
| | Apple-M | 55.9 (±7.0) | 54.9 (±0.4) | 48.3 (±7.5) | 64.3 (±1.2) | 77.8 (±3.5) |
| | LSM | 52.9 (±5.6) | 55.0 (±1.0) | 45.8 (±7.4) | 64.8 (±1.3) | 79.0 (±4.7) |
| 100% | *xMAE* | 68.8 (±4.8) | 58.5 (±1.1) | 65.1 (±12.5) | 66.4 (±2.3) | 81.4 (±5.1) |
| | Apple-M | 56.3 (±6.7) | 56.3 (±1.2) | 49.4 (±7.3) | 65.2 (±1.7) | 80.7 (±4.6) |
| | LSM | 53.8 (±5.6) | 55.0 (±0.9) | 47.3 (±7.8) | 64.7 (±1.6) | 78.8 (±5.4) |

that *xMAE* learns transferable physiological structure between ECG and PPG, rather than overfitting to a specific PPG sensing location. Fingertip PPG may provide cleaner signals during pretraining, which could explain its slightly stronger performance in several tasks. However, matched-domain wrist pretraining does not change the overall conclusion: the physiological structure learned by *xMAE* transfers well to wrist-based wearable health tasks.

### H.3. Data Efficiency Analysis

To evaluate whether *xMAE* remains effective in low-data pretraining regimes, we compare *xMAE* with Apple-M and LSM under different pretraining volumes: 3%, 30%, and 100%. Table 15 reports AUROC scores across five downstream tasks.

The results show that *xMAE*'s advantage is broad across different pretraining volumes. Even with only 3% of the pretraining data, *xMAE* outperforms the baselines on most tasks, especially hypertension in the lab setting, A1C, and PVC detection. As the pretraining volume increases from 3% to 100%, *xMAE* generally improves further and maintains strong performance across tasks. These results suggest that the physiology-aware inductive bias in *xMAE* is especially beneficial in low-data regimes and generalizes across downstream wearable health tasks.

### H.4. Linear Probing vs. Fine-Tuning

Our primary goal is to evaluate the quality of the learned representations independent of task-specific adaptation, which is why we emphasize linear probing in the main evaluation. However, fine-tuning provides a complementary view of how much downstream performance can be improved when the full model is adapted with task-specific supervision. We therefore compare *xMAE* under both settings: *xMAE*-FT denotes fine-tuning, and *xMAE*-LP denotes linear probing.

As shown in Table 16, fine-tuning generally improves performance on several tasks, including PVC detection, ectopic beats detection, and sleep staging. Linear probing remains competitive on other tasks, such as hypertension and several lab-test prediction tasks. This suggests that the learned representation is informative even without task-specific adaptation, while still benefiting from fine-tuning when additional supervision is available.

We also observe that performance varies across tasks, partly due to differences in effective data scale. Some tasks contain fewer segments per subject, which can reduce the amount of training signal and lead to higher variance. Overall, we view linear probing and fine-tuning as complementary: linear probing evaluates representation quality in a controlled setting, while fine-tuning reflects the best achievable downstream task performance.

## I. Additional Ablation Study

To study the effectiveness of our main modules (continuous masking and directional cross-attention) on encouraging *xMAE* encoding of physiologically meaningful timing features, we set up baselines as follows:

**Setup for Baseline1 (w/o Directional Cross-Attention)** We create a baseline where the masking strategy is kept the same as *xMAE*, yet the directional cross-attention is replaced with a simple concatenation operation.

**Setup for Baseline2 (w/o Continuous Masking)** We create another baseline with the same architecture as *xMAE*, but

**Table 16.** Classification comparison between fine-tuning and linear probing for *xMAE*. Results are reported as AUROC.

| Task | *xMAE*-FT | *xMAE*-LP |
|---|---|---|
| Hypertension (lab) | 67.0 (±9.7) | 68.8 (±4.8) |
| Hypertension (free-living) | 61.4 (±0.9) | 58.5 (±1.1) |
| PVC | 88.8 (±3.7) | 81.4 (±5.1) |
| Ectopic Beats | 93.8 (±1.1) | 87.8 (±2.3) |
| A1C | 60.5 (±15.1) | 65.1 (±12.5) |
| Hemoglobin | 60.5 (±18.7) | 62.0 (±16.0) |
| Platelets | 69.0 (±15.8) | 68.6 (±16.5) |
| Sodium | 63.7 (±17.2) | 61.7 (±16.1) |
| Wake | 71.3 (±1.7) | 66.4 (±2.3) |
| Light | 58.5 (±0.9) | 57.5 (±1.3) |
| Deep | 60.7 (±3.2) | 55.9 (±5.3) |
| REM | 55.3 (±3.2) | 54.5 (±2.5) |

replacing with random masks for ECG.

**Setup for Baseline3 (ECG Masking Ratio 0.95)**     We create this baseline with the same architecture as *xMAE*, but with ECG masking ratio of 0.95.

In inference, we follow the same masking strategy for ECG as defined during pretraining and leave PPG unmasked, and the reconstructed ECG is used to calculate error with the ground truth ECG.

**Analysis**     We utilize *AFib* as the validation dataset in this experiment. Figure 17 depicts the errors of *xMAE* and baselines that remove certain components as described above. Notably, cross-attention plays an important role in encouraging models to infer from PPG. In contrast, random masking or excessively high masking (i.e., 95% of ECG segments) substantially degrades ECG reconstruction quality, since the remaining visible ECG fragments no longer preserve sufficient temporal context to reveal the ECG–PPG relationship, thereby breaking the intended inductive bias. Figure 18 provides a number of examples. Overall, we believe all components contribute to the final performance of *xMAE*.

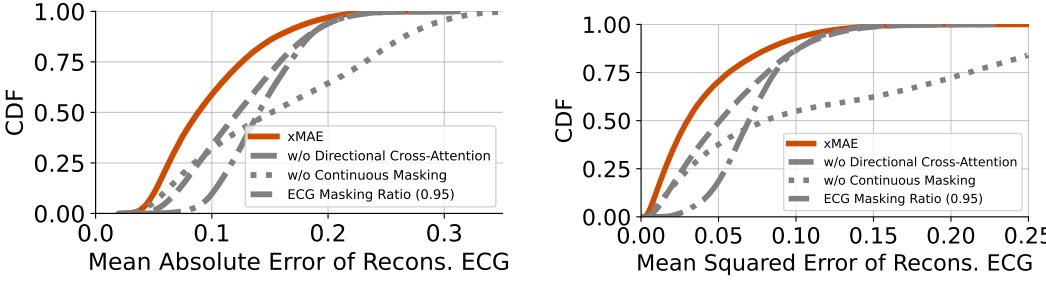

**Figure 17.** ECG reconstruction errors between *xMAE* and ablation baselines. (Left) Mean Absolute Error. (Right) Mean Squared Error.

Table 17 depicts the other design choices in *xMAE*: mask ratios and patch sizes. Overall, masking 90% of each ECG segment with a patch size of 40 yields the best combination of results.

## J. Case Study: Physiological Fidelity of Reconstructed ECG

**Dataset and Preprocessing** We leverage Samsung-sponsored dataset which has 30-s spot-check ECG sampled at 500 Hz and PPG sampled at 100 Hz. Briefly, We perform the same preprocessing procedures as defined in Appendix C, and

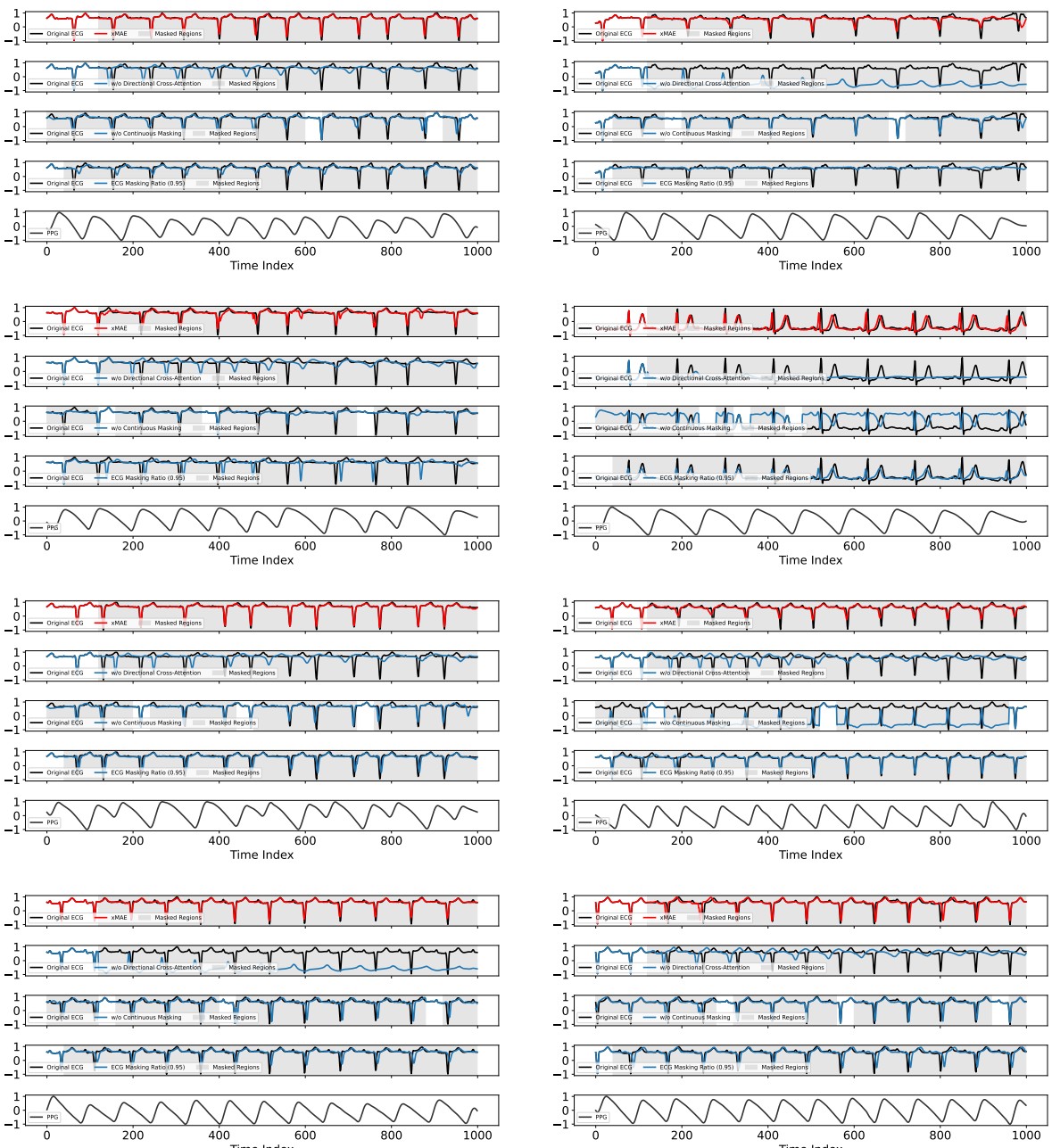

**Figure 18.** Ablation study of design choices by ECG reconstruction.

**Table 17.** Additional Ablation Study. The default settings are in gray area.

|  | Hypertension (lab) | Ectopic Beats |
|---|---|---|
| 60% | 65.6 (±6.5) | 86.8 (±1.7) |
| 70% | 63.7 (±5.5) | 84.6 (±1.9) |
| 80% | 64.2 (±5.6) | 85.3 (±3.8) |
| 90% | **68.8** (±4.8) | **87.8** (±2.3) |

**(a) Mask Ratio**

|  | Hypertension (lab) | Ectopic Beats |
|---|---|---|
| 10 | 62.4 (±6.0) | **88.6** (±1.5) |
| 20 | 67.2 (±5.4) | 87.9 (±2.0) |
| 40 | **68.8** (±4.8) | 87.8 (±2.3) |
| 100 | 64.1 (±5.7) | 80.5 (±2.5) |

**(b) Patch Size (patch sizes are selected because they are divisible by 1000).**

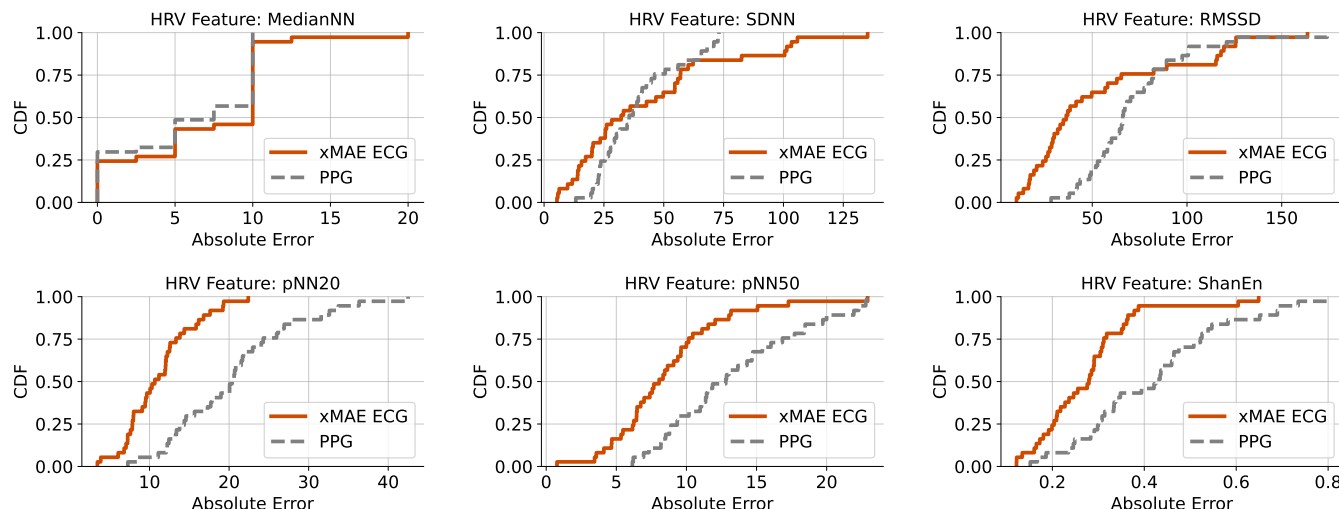

**Figure 19. Evaluating ECG reconstruction quality via HRV features.** CDFs of absolute error for HRV metrics computed from *xMAE*-reconstructed ECG and from PPG signals. Across all features, *xMAE* exhibits consistently lower error, indicating improved preservation of beat-to-beat timing and temporal structure by capturing physiologically meaningful ECG dynamics.

down-sample the signals to 100 Hz for evaluation. As a result, the evaluation is left with 2.6k 10-s segments from 38 subjects. When calculating HRV features, we concatenate the signals back to 30-second segments for a stable estimation of HRV features.

**ECG Reconstruction Steps** Owing to our masking method and pretraining objective, *xMAE* has the ability to reconstruct ECG given continuous PPG. Yet, *xMAE* still needs a visible ECG signal as input. We perform the following operations to fulfill this task. From one subject, we first held-out a good quality ECG template (e.g., 1.2 seconds long) and the corresponding PPG. When reconstructing ECG with newly coming PPG segments, we concatenate the PPG segment in the held-out pair to the incoming PPG to make it a 10-s PPG. We concatenate PPG by its valleys, and apply a smooth filter for signal continuity. Lastly, we pass the signals to *xMAE* for ECG reconstruction.

**ECG Reconstruction based HRV** We evaluate the physiological fidelity of reconstructed ECG by comparing HRV features computed from reconstructed ECG and ground truth ECG. We utilize the HRV features from the accompanying PPG as a baseline. We utilize Neurokit2 (Makowski et al., 2021), a state-of-the-art biosignal processing Python toolkit, for peak detection for both signals and calculate HRV features. We list the features that are widely utilized in health applications and their meanings available as follows:

- **MedianNN**: Median NN is a time-domain measure representing the median of all normal-to-normal (NN) heart beat intervals (the time between consecutive heartbeats) recorded over a specific period.

- **SDNN**: SDNN (Standard Deviation of Normal-to-Normal intervals) measures the overall variation between the heartbeats over a period, reflecting the body's total adaptability to stress and recovery; a higher SDNN generally means better resilience and a stronger autonomic nervous system, while a lower SDNN suggests higher stress or poor recovery.

- **RMSSD**: RMSSD (Root Mean Square of Successive Differences) is the metric that measures the milliseconds of variation between consecutive heartbeats, reflecting short-term parasympathetic nervous system activity (rest and digest) and providing insights into recovery and stress, with higher numbers generally indicating better resilience and readiness.

- **pNN20**: pNN20 is the metric representing the percentage of time successive heartbeats (RR intervals) differ by more than 20 milliseconds, indicating rapid, short-term adjustments reflecting parasympathetic (rest-and-digest) activity, useful for assessing autonomic nervous system balance, stress, and recovery.

- **pNN50**: pNN50 (percentage of NN intervals >50ms) is the measure showing the percentage of consecutive heartbeats that differ by over 50 milliseconds, reflecting strong parasympathetic nervous system (PNS) activity (rest-and-digest) and rapid heart rate adjustments, indicating good autonomic balance and cardiovascular health, with higher values generally signaling better fitness and resilience.

- **ShanEn**: ShanEn (Shannon entropy) is a non-linear measurement used in heart rate variability analysis to quantify the complexity and unpredictability of the beat-to-beat heart rhythm time series. It is derived from information theory and assesses the distribution of heart interbeat intervals (RRI). A higher ShanEn value generally indicates greater flexibility and resilience in the body's autonomic control of the heart, while a lower value may point to increased stress, fatigue, or potential health issues.

**Results and Analysis**     Figure 19 shows the overall comparisons between HRV features derived from reconstructed ECG of *xMAE* and PPG signals where *xMAE* based HRV features has lower errors, suggesting that *xMAE* encodes meaningful ECG dynamics in its latent PPG space (e.g., timing between PPG and ECG signals). MedianNN and SDNN primarily capture longer-timescale variability in heart rate over the recording window, making them more sensitive to slow trends, baseline drift, and low-frequency noise. As a result, their error distributions are comparable in our setting, since these features are computed from 30-second segments, reflecting the limitation that our wearable dataset contains only 30-second spot-check ECG recordings. In contrast, RMSSD and pNN metrics emphasize short-term, beat-to-beat variability, focusing on rapid fluctuations between successive heartbeats. These features are therefore better suited to short recording windows and benefit more directly from improvements in beat-level temporal modeling. We believe *xMAE* captures the fast electrical-to-mechanical timing between ECG and PPG relatively well. Figure 20 and Figure 21 present a few of the reconstructed ECGs from *xMAE* with peaks labeled as blue dots. Notably, R peaks detected from *xMAE* reconstructed ECG are aligned well with the ground truth ECGs. The last few subplots in Figure 21 also demonstrate some failure cases where the amplitudes of T waves in reconstructed ECGs are erroneous, resulting in incorrect peak detections. As discussed, we plan to modify the loss function in *xMAE* to focus on more timing-related information from ECG's P waves, QRS complex, etc. Overall, we believe the HRV features calculated from *xMAE* reconstructed ECG are proving that *xMAE* encodes timing information across signals, and fusing the HRV features calculated from *xMAE* reconstructed ECG with that of PPG signals could boost the performance of numerous health applications such as stress monitoring and sleep.

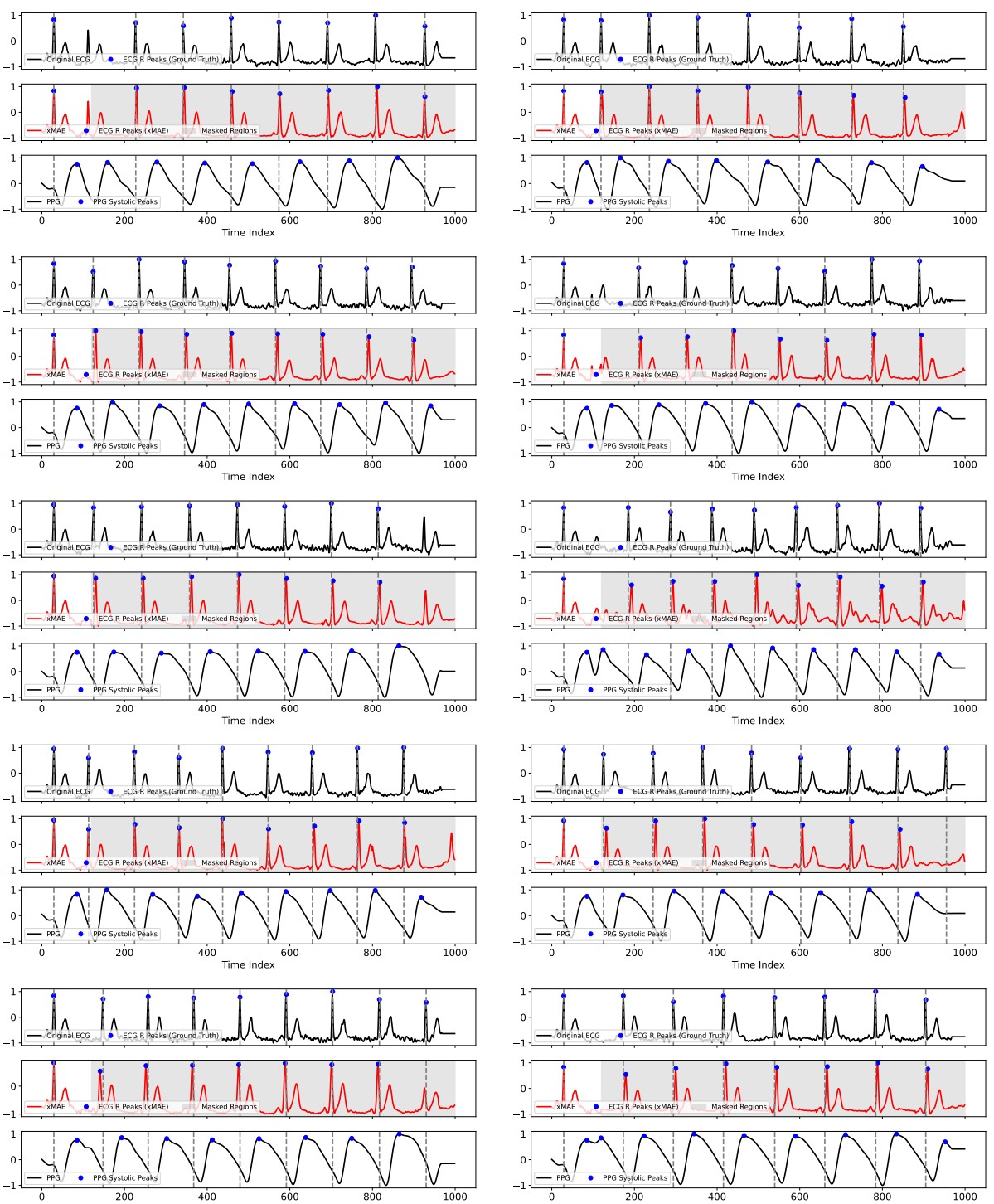

**Figure 20.** ECG reconstruction illustration 1 for HRV.

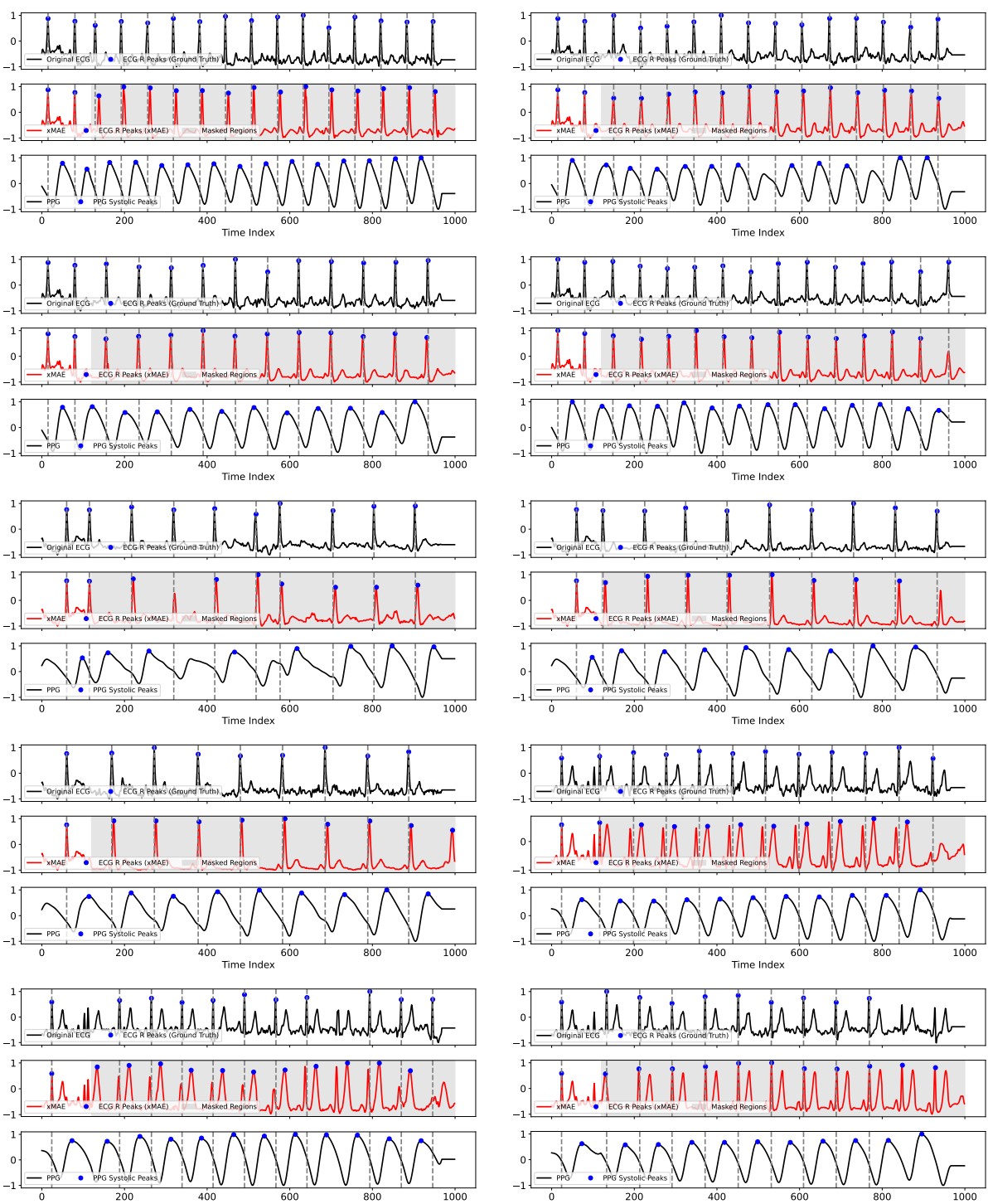

**Figure 21.** ECG reconstruction illustration 2 for HRV (with erroneous cases).

# K. Visualization of PPG Embeddings in Downstream Tasks

We provide 2D visualizations of PPG embeddings learned by *xMAE* under both linear-probing and fine-tuning protocols using t-SNE dimensionality reduction (Maaten & Hinton, 2008). As shown in Figure 22, Figure 23, and Figure 24, embeddings obtained from linear probing already exhibit meaningful class structure, with samples from different clinical categories forming separable clusters. This suggests that *xMAE* learns task-relevant features during pretraining that are readily accessible to simple linear classifiers, providing qualitative evidence of the transferability of the learned representations across diverse downstream classification tasks.

After fine-tuning, clusters appear tighter. This increased compactness indicates reduced intra-class variance and improved consistency of the learned embeddings, reflecting better alignment between the representation space and downstream task objectives. Importantly, these changes suggest that fine-tuning refines the embedding geometry in a way that enhances both classification accuracy and representation reliability. Together, these visualizations illustrate how *xMAE* produces structured and adaptable PPG representations that remain semantically meaningful under linear evaluation while benefiting from further task-specific refinement.

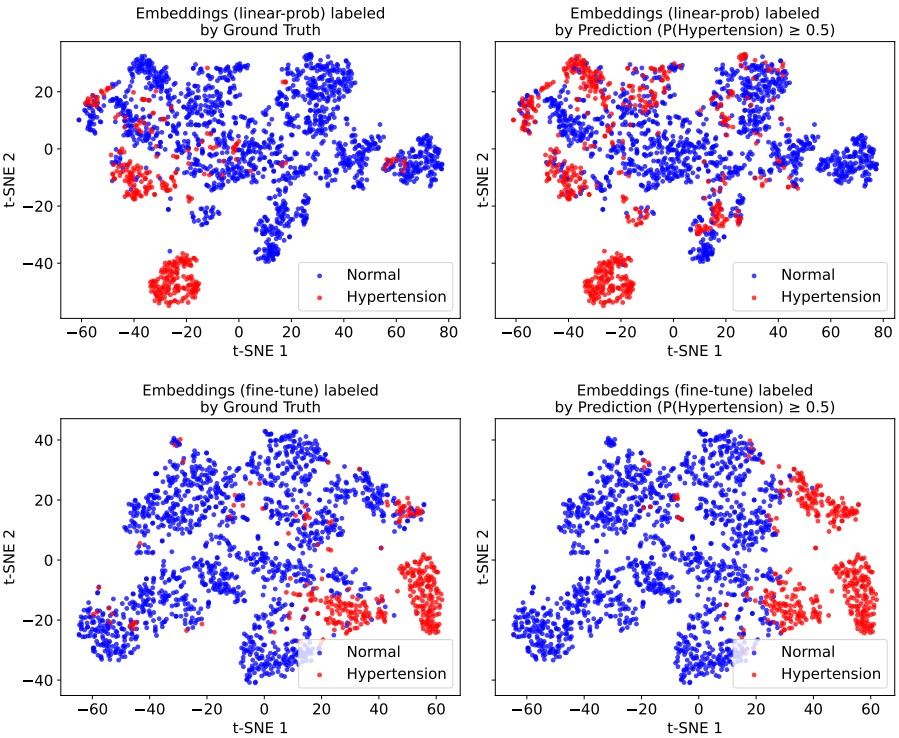

**Figure 22.** t-SNE plots of PPG embeddings from *Hypertension (lab)* task under linear-probing and finetuning.

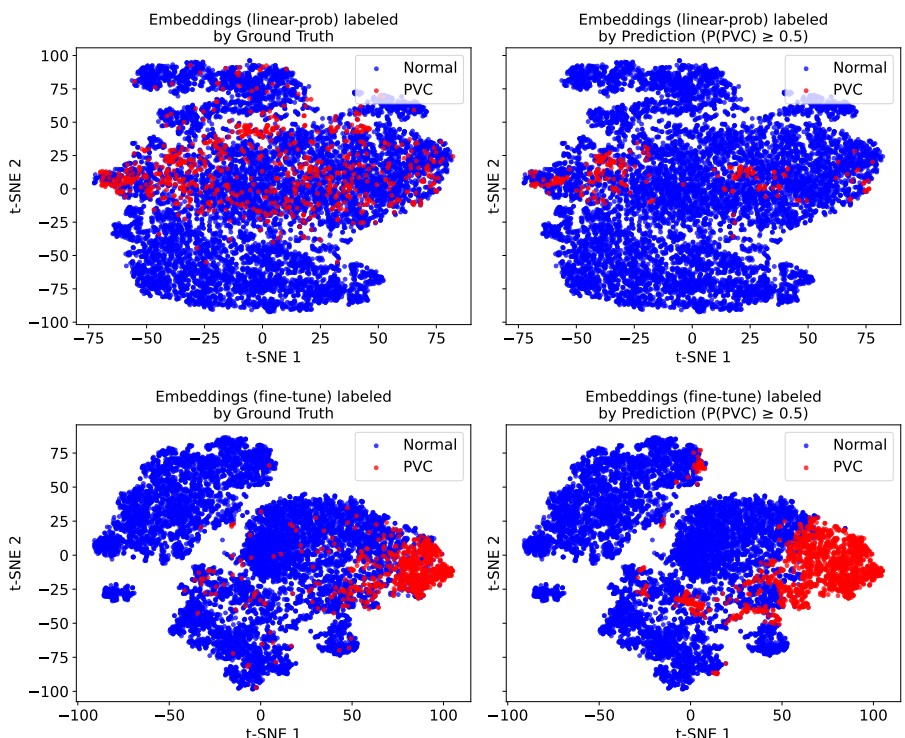

**Figure 23.** t-SNE plots of PPG embeddings from *PVC* task under linear-probing and finetuning.

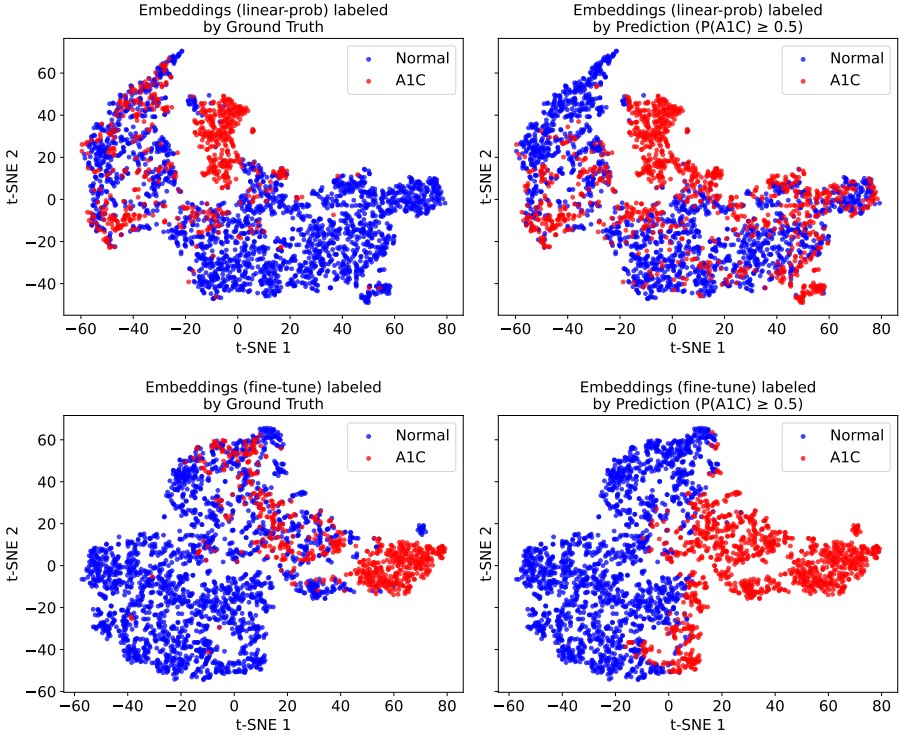

**Figure 24.** t-SNE plots of PPG embeddings from *A1C* task under linear-probing and finetuning.

# L. Computational Cost Analysis

We compare the computational cost of *xMAE* against representative open-source biosignal foundation models, including PulsePPG, AnyPPG, and PaPaGei. We report three complementary metrics: the number of trainable parameters, theoretical computational complexity measured in GFLOPs, and empirical inference throughput measured in segments per second. All experiments are performed on an NVIDIA H200. See code below.

As shown in Table 18, PulsePPG exhibits the highest computational cost, with 28.5M parameters and 28.5 GFLOPs per forward pass, resulting in a relatively low throughput of 3.4k segments/sec. In contrast, AnyPPG and PaPaGei are substantially more lightweight, requiring fewer than 6M parameters and under 0.2 GFLOPs, which enables significantly higher throughput (22k and 33k segments/sec, respectively). However, these efficiency gains come at the expense of representational capacity, as reflected in their downstream performance (Table 4). *xMAE* occupies a favorable middle ground between these extremes. With 6.5M parameters and 0.165 GFLOPs per forward pass, *xMAE* remains comparable in complexity to lightweight baselines while achieving a throughput of 24k segments/sec, over $7\times$ faster than PulsePPG, without sacrificing modeling expressiveness.

Overall, these results indicate that *xMAE* achieves a strong efficiency–performance trade-off, delivering competitive computational efficiency while retaining sufficient capacity for robust downstream transfer. This makes *xMAE* particularly well-suited for large-scale pretraining and deployment in resource-constrained wearables.

| Model | # Params (M) | GFLOPs | Throughput (k segments/sec) |
|---|---|---|---|
| PulsePPG (Saha et al., 2025) | 28.5 | 28.5 | 3.4 |
| AnyPPG (Nie et al., 2025) | 5.8 | 0.194 | 22 |
| PaPaGei (Pillai et al., 2024) | 5.7 | 0.0598 | 33 |
| *xMAE* | 6.5 | 0.165 | 24 |

**Table 18.** Computational cost comparisons with open-source models.

**Calculating FLOPs**

```
@torch.no_grad()
def get_macs_flops(model, L=1000, device="cuda"):
    model.eval().to(device)
    x = torch.randn(1, L, device=device)
    flops = FlopCountAnalysis(model, (x,))
    return flops.total()
```

**Calculate Throughput**

```
@torch.no_grad()
def measure_throughput(model, B=128, L=1000, iters=300, warmup=50, device="cuda"):
    model.eval().to(device)
    x = torch.randn(B, L, device=device)
    for _ in range(warmup):
        _ = model(x)
    torch.cuda.synchronize()
    start = torch.cuda.Event(True)
    end = torch.cuda.Event(True)
    start.record()
    for _ in range(iters):
        _ = model(x)
    end.record()
    torch.cuda.synchronize()
    total_ms = start.elapsed_time(end)
    return (iters * B) / (total_ms / 1000)
```

