# OpenReview forum: "Physiology-Aware Masked Cross-Modal Reconstruction for Biosignal Representation Learning"
_ICML.cc/2026/Conference — ICML 2026 regular_

### Official Review · Reviewer_QZ9Z · 2026-02-26

**Soundness:** 3
**Presentation:** 4
**Significance:** 4
**Originality:** 3
**Overall Recommendation:** 5
**Confidence:** 4

**Summary:**

The paper introduces a framework, designated xMAE, based on asymmetric masked modeling of coupled data, where a cross-attention mechanism reconstructs masked segments by primarily leveraging cross-modal relationships. This masked modeling follows a curriculum-based strategy that encourages the model to rely increasingly on the paired (unmasked) modality for reconstruction.

The work focuses on the coupled ECG+PPG case, masking the former at progressively higher rates to exploit the cross-modal dependencies with PPG during reconstruction. This pre-training yields robust PPG representations which, in most instances, provide superior embeddings for linear probing compared to those produced by other unimodal and multimodal baseline objectives. Furthermore, the method remains competitive with established open-source large-scale pretrained models in the field. Through a series of analyses, the authors also demonstrate that xMAE’s reconstruction of masked ECG signals is substantially more accurate than that of existing baseline methods.

**Compliance With Llm Reviewing Policy:**

Affirmed.

**Final Justification:**

The current submission presents a compelling framework and contribution to multimodal biosignal modeling. Furthermore, the evaluation is robust and substantial. It warrants acceptance.

**Key Questions For Authors:**

1) While the framework is presented as a general approach for multimodal physiological signals, I suspect the optimal curriculum may be highly modality-dependent. A strategy designed for ECG/PPG may not directly translate to other biosignal pairs. I would be interested in the authors' perspective on this potential modality-specific sensitivity. If the authors agree that the curriculum requirements might vary significantly across different physiological domains, I would argue that a brief discussion of this for future research would be a highly relevant addition to the final version of the paper.

2) Given that the pre-training data originates from ICU settings, there is a likely skew toward specific demographics / settings (e.g., older populations, diseased states). Crucially, please clarify if the (redacted) evaluation datasets align more closely with the ICU pre-training environment than the data used for the baseline Foundation Models (FMs). If such an alignment exists, it might suggest that the performance gains are driven by data distribution overlap rather than the methodological pipeline. Moreover, a discussion or disclaimer, particularly regarding the limitations in clinical deployment, given the delicate domain, for broader populations would be a valuable addition.


3) Could the authors clarify whether the reported statistical tests for the results obtained have been corrected for multiple comparisons? If not, this should be done.

4) The manuscript attributes the downside of existing methods to the temporal lag between modalities. However, an alternative hypothesis is that joint masked modeling (as seen in Fang et al.) leads to sub-optimal representations because the model attempts to optimize multiple modalities simultaneously. Is it possible that xMAE’s success stems from using the full context of a second modality to aid a single-modality optimization process (i.e., avoiding the simultaneous reconstruction of multiple modalities at the same time), rather than specifically "solving" a temporal offset? I would appreciate the authors’ thoughts on whether the benefit is derived from this asymmetric optimization structure rather than the temporal dynamics themselves. If not, why?


5) The text states that xMAE does not rely on modality alignment. However, by leveraging one modality to reconstruct the other, the framework appears to inherently perform a form of alignment, simply one that lacks the rigid, sample-level temporal constraints of some prior work. Would the authors agree that this is a "latent" or "relaxed" alignment rather than a total absence of it?

The current submission presents a compelling framework and, in my view, already warrants acceptance. However, I believe the framing of certain architectural advantages requires further nuance. Provided the authors can offer principled answers to the points above, I would be inclined to upgrade my recommendation to a Strong Accept.

**Limitations:**

Mostly yes, however, particularly given the delicate nature of some possible future work in the clinical domain, I'd reinforce the notion that this is early work which must be carefully validated by future research.

**Strengths And Weaknesses:**

Following are presented the encountered strengths and weaknesses and a couple of notes for the proof version.

Strengths:
- The manuscript addresses a highly relevant challenge: the development of multimodal foundation models within the healthcare domain.
- The results are promising, demonstrating performance that is competitive with established pre-training objectives and existing open-source PPG foundation models.
- Although the study focuses on ECG/PPG coupling, the authors correctly identify the framework's potential as a broader contribution to multimodal physiological sensing.
- For the most part, the paper is well-written and logically organized.
- The inclusion of substantial supplemental information and ablation studies effectively contextualizes and substantiates the primary results.
- The evaluation is robust, spanning both controlled in-lab environments and ecological free-living conditions.


Weaknesses:
- While generally clear, certain experimental details are unintuitive. Specifically, it is not immediately clear that models in Tables 1 and 2 were re-trained (e.g., the PaPaGei model), whereas Table 3 utilizes original released weights. If this is the case, explicitly stating that re-trained versions using the original pipeline/objectives on the used pre-traned dataset would resolve initial reader confusion.
- The masking curriculum is quite aggressive. While this is effective for signals with strong temporal self-correlation, it may pose challenges for other biosignal pairs. If the framework is intended for "any pair of biosignals," this limitation should be discussed to guide future research.
- Neither the model weights nor the evaluation datasets are slated for public release. While the necessity of data privacy is understood, this lack of transparency may hinder future benchmarking and reproducibility.

A couple of notes to adjust for the proof version (if desired): SleepFM reference is outdated, the term “reasoning” between modalities seems to be overloaded given that the term is used for other sub-fields in quite a different way; it might result confusing.

---

> ### Author Rebuttal · Authors · 2026-03-29
>
> We thank Reviewer QZ9Z for the insightful feedback and for recognizing our contributions to multimodal healthcare models. We appreciate the acknowledgment of our robust evaluation across both in-lab and free-living settings, as well as the broader potential of our framework. We are also glad that the paper organization and supplementary materials were helpful. We address the comments below.
>
>
> > Modality-Specific Sensitivity of the Curriculum
>
> We thank the reviewer for this insightful observation. We agree that the optimal curriculum is modality-dependent. In our case, strong electromechanical coupling between ECG and PPG enables effective learning even with high masking (90%). For weaker modality pairs (e.g., EMG and motion), lower masking or a more gradual curriculum may be needed. We also note that curriculum learning may not always be necessary, but for ECG–PPG, it improves stability and supports hierarchical representation learning. We will clarify this limitation and highlight adapting curricula across modalities as future work.
>
>
>
> > Data Distribution and Population Skew
>
> This is an important point. Our evaluation datasets differ substantially from the ICU pre-training data, spanning lab and free-living settings and using wrist PPG, whereas ICU data typically uses fingertip sensors. This creates a clear domain gap in noise and hardware, suggesting that performance gains come from the physiological inductive bias rather than data overlap.
>
> We also acknowledge the demographic skew of ICU data toward older populations, which limits generalizability. We will include a clear disclaimer and discuss this limitation in the revision.
>
>
>
> > Asymmetric Optimization vs. Temporal Dynamics
>
> We thank the reviewer for this profound and insightful question. We agree that asymmetric optimization likely contributes to the performance gains. In our view, these two perspectives are not mutually exclusive, but fundamentally connected.
>
> In symmetric baselines where both modalities are randomly masked and reconstructed (e.g., LSM, pretrained on our data), the model often faces a “double-blind” optimization problem. Because key physiological events in ECG and PPG are temporally coupled, random masking can remove corresponding features from both signals simultaneously, reducing the mutual information needed to learn their relationship. This can lead to suboptimal representations.
>
> xMAE reconstructs ECG using the full PPG context, providing a stable physiological anchor. This asymmetric structure allows the model to focus on learning the mapping between mechanical pulses and electrical activity. We believe this design is an important driver of the improved performance.
>
> Thus, asymmetric optimization aligns with asymmetric temporal observability rather than being an independent factor.
>
> To further disentangle these effects, evaluating under perfectly synchronized or synthetic settings would be informative. If the advantage persists, it suggests optimization dominates; if it diminishes, temporal dynamics play a key role. We agree this is an important direction for future study and will include this discussion in the manuscript.
>
>
>
> > Latent vs. Total Absence of Alignment
>
> We thank the reviewer for this insightful observation. We agree that xMAE does not eliminate alignment, but rather shifts it from rigid sample-level synchronization to a more physiologically grounded form.
>
> In our setting, ECG and PPG are temporally aligned at the sequence level. However, instead of enforcing strict sample-level correspondence, xMAE performs a form of event-level alignment, which can be viewed as a physiologically motivated relaxed alignment. This enables the model to capture more meaningful correspondences (such as the dynamic relationship between electrical activation and pulse arrival) that are not well represented by rigid timestamp matching.
>
>
>
> > Addressing Other Comments
>
> - We performed t-tests comparing xMAE with the top baseline for each task and applied the Benjamini–Hochberg procedure for correction. Most tasks remain significant at $p < 0.05$ after correction. We will update this in the manuscript.
> - We will make the description of the baseline setup clear in the tables: Tables 1–2 baselines are re-trained on the same dataset as xMAE, and Table 3 uses the original released weights for zero-shot comparison.
> - We acknowledge concerns about reproducibility. Due to data privacy, we cannot release raw datasets or pretrained weights. With fully released training code and public datasets (MIMIC-III and DREAMT), some of our results can be reproduced.
> - We will update the SleepFM reference and refine the use of “reasoning” to avoid confusion.
>
> ---
> We sincerely thank Reviewer QZ9Z for the thoughtful feedback. If the clarifications above are helpful, we would greatly appreciate reconsideration of the score. We are happy to address any further questions.

---

> > ### Author Rebuttal · Reviewer_QZ9Z · 2026-04-02
> >
> > The authors answered appropriately to most of my questions.
> >
> > I maintain my score of 5, as I believe the manuscript, also in light of the proposed additions, clearly warrants acceptance. My remaining minor concerns pertain to reproducibility, which cannot be solved, and the generalizability of the findings to other modality pairs.

---

> > > ### Author Response · Authors · 2026-04-04
> > >
> > > We thank Reviewer QZ9Z for the acknowledgement and are glad our rebuttal addressed your concerns. We truly appreciate your time and your recognition of our work. We will incorporate your feedback to further strengthen the paper.

---

### Official Review · Reviewer_3VLY · 2026-03-04

**Soundness:** 3
**Presentation:** 3
**Significance:** 2
**Originality:** 2
**Overall Recommendation:** 4
**Confidence:** 3

**Summary:**

The paper introduces xMAE, a self-supervised representation learning framework for biosignals that shifts the multimodal pretraining paradigm from treating signals as interchangeable views to modeling them as temporally ordered stages of a shared physiological process. By leveraging the "asymmetric temporal observability" between ECG (the electrical trigger) and PPG (the delayed mechanical response), the authors implement a masked cross-modal reconstruction objective using a curriculum masking strategy and directional cross-attention. This approach forces the PPG encoder to capture deep cardiovascular dynamics and timing structures—such as pulse arrival time—that are often ignored by symmetric models. The resulting representations significantly outperform state-of-the-art baselines across 15 of 19 downstream clinical and demographic tasks, even when only the single, ubiquitous PPG modality is available at inference time.

**Compliance With Llm Reviewing Policy:**

Affirmed.

**Key Questions For Authors:**

1. **Scalability with Unpaired Data**: The current framework relies strictly on synchronized ECG-PPG pairs for pretraining. Given that unpaired PPG data is far more abundant in wearable contexts, have the authors considered or tested a semi-supervised variant where the model also learns from unpaired PPG signals? Understanding the model's performance when the "ECG scaffold" is unavailable for a portion of the data would clarify its true scalability and practical utility.

2. **Impact of the MSE Loss Function**: The authors acknowledge that MSE may not capture fine-grained ECG features like P-waves. Could the authors provide evidence or discussion on whether a perception-based loss or a coordinate-based loss for specific physiological landmarks (e.g., QRS complex detection) was tested? If a more specialized loss does not significantly improve the downstream PPG representation, it would strengthen the claim that global timing and pulse arrival dynamics are the primary drivers of the observed performance gains.

3. **Curriculum Strategy Sensitivity**: The curriculum strategy increases the masking ratio based on a 10% relative improvement threshold. How sensitive is the final downstream performance to this specific threshold and the step size of 5%? Clarifying whether the model's success is highly dependent on this specific hyperparameter tuning—or if a fixed high masking ratio could achieve comparable results with longer training—would help assess the framework's robustness and ease of application to other biosignal pairs.

**Limitations:**

yes

**Strengths And Weaknesses:**

**Strengths**

* **Effective Physiological Inductive Bias**: The work successfully translates the "electrical-trigger-to-mechanical-response" directional relationship between ECG and PPG into a pretraining constraint. This demonstrates that incorporating asymmetric temporal observability is more effective for capturing core cardiovascular features than generic symmetric multimodal alignment.
* **Significant Clinical and Practical Utility**: Despite being trained on clinical data, the model demonstrates robust generalization across 15 of 19 downstream tasks—including hypertension and arrhythmia detection—using only single-modality PPG at inference. This highlights its high potential for real-world deployment on resource-constrained wearable devices.

**Weaknesses**

* **Heavy Dependency on Paired Data**: The pretraining phase strictly requires high-quality, synchronized ECG-PPG recordings. This limitation hinders the model's ability to leverage the vast amounts of unpaired or unlabelled wearable data available in the wild.
* **Limitations of the Loss Function**: The reliance on a simple Mean Squared Error (MSE) loss may adequately capture dominant R-peak timing but potentially overlooks finer-grained morphological features, such as P-waves or T-wave characteristics, which are critical for comprehensive cardiac diagnostics.
* **Insufficient Latent Space Interpretability**: While the results show improved class separation in downstream tasks, there is a lack of deep qualitative analysis regarding exactly which physiological features (beyond pulse arrival timing) are being encoded within the PPG latent space.

---

> ### Author Rebuttal · Authors · 2026-03-29
>
> We thank Reviewer 3VLY for the insightful and constructive feedback. We are encouraged that the reviewer recognized our contributions in formalizing the physiological inductive bias and the strong performance across diverse downstream tasks for real-world deployment. We address the reviewer’s comments below.
>
>
> > Scalability with Unpaired Data
>
> We agree that reliance on paired ECG–PPG data is a constraint. However, we believe such a constraint can be relaxed in a practical semi-supervised pipeline. We envision a two-stage training strategy: (1) paired pretraining to learn physiological structure (e.g., electromechanical coupling), and (2) scaling with large unpaired PPG using self-supervised objectives. This reflects realistic data availability and improves robustness and generalization. We believe this is a natural extension of xMAE and will clarify it as a promising direction for future work.
>
>
> > Insufficient Latent Space Interpretability
>
> We agree that better understanding what physiological features are encoded is important. To address this question, we performed morphology feature comparisons on ground truth ECG and reconstructed ECG from xMAE, using neurokit2 ECG processing tools. We calculated 5 features, including 1) QRS width, 2) PR Interval, 3) QT Interval, 4) ST level, and 5) ST slope.
>
> |                | QRS width (ms) | PR interval (ms) | QT interval (ms) | ST level | ST slope |
> |----------------|--------------|----------------|----------------|----------|----------|
> | GT ECG (mean)    | 162.25       | 147.71         | 506.84         | -0.04    | 4.17     |
> | xMAE ECG (mean)  | 190.62       | 129.26         | 529.59         | -0.07    | 2.62     |
> | Diff. (mean)     | 28.37        | -18.45         | 22.75          | -0.03    | -1.55    |
> | GT ECG (median)  | 130.00       | 160.00         | 470.00         | -0.01    | 2.77     |
> | xMAE ECG (median)| 140.00       | 160.00         | 490.00         | -0.08    | 1.46     |
> | Diff. (median)   | 10.00        | 0.00           | 20.00          | -0.07    | -1.31    |
>
> As shown in the table above, xMAE preserves key features with small median errors, including QRS width (10 ms), PR interval (0 ms), and QT interval (20 ms). These features reflect intra-beat structure that cannot be obtained from HR or HRV alone, indicating that the model captures more than timing information. We also observe larger deviations in ST-segment characteristics. This is expected, since ST features reflect ventricular repolarization, which has weak or no direct analog in PPG. In addition, these features are low-amplitude and sensitive to baseline shifts, making them inherently more difficult to reconstruct.
>
> Overall, these results demonstrate that xMAE also captures morphological features by leveraging shared structure, such as electromechanical coupling, enabling richer representations. We will add this analysis into the manuscript to enhance the interpretability of the latent space.
>
> > Impact of Loss Function
>
> We thank the reviewer for this thoughtful question. We use MSE as a simple and stable objective, since our goal is not high-fidelity ECG reconstruction but to use ECG as supervision to shape PPG representations for real-world deployment. While MSE may underemphasize fine structures (e.g., ST segment), it encourages learning physiologically inferable structure rather than exact reconstruction.
>
> We view MSE as a soft constraint that avoids hallucinating non-inferable details. As shown above, key features (QRS, PR) are preserved, while weakly coupled ones (ST) show larger deviations, consistent with our design.
>
> Overall, while we did not explicitly evaluate perception-based or landmark-aware losses, our results suggest that even with MSE, the model already captures both timing and partial morphology. We expect that more specialized losses may improve ECG reconstruction fidelity, but may not substantially change downstream PPG performance, since features with weak or no physiological coupling to PPG are inherently difficult to infer. We will clarify this point and include it as an important direction for future work.
>
>
> > Curriculum Strategy Sensitivity
>
> We should have made it clear. We evaluated this in Table 7. Both strategies (with or without curriculum strategy) achieve comparable performance, with curriculum masking providing slightly more consistent results (lower variance), suggesting that xMAE is not highly sensitive to the specific threshold (10%) or step size (5%). This also indicates that the gains are driven by the inductive bias rather than specific curriculum design choices, making the framework robust and easy to apply to other biosignals. We will revise the manuscript to clarify this point.
>
> ---
> We sincerely thank Reviewer 3VLY for the thoughtful feedback and for taking the time to engage with our work. If the clarifications above are helpful, we would greatly appreciate reconsideration of the score. We are also happy to address any further questions.

---

> > ### Author Rebuttal · Reviewer_3VLY · 2026-04-01
> >
> > The author's response addressed some of my concerns; I am maintaining my score of four.

---

> > > ### Author Response · Authors · 2026-04-04
> > >
> > > We thank Reviewer 3VLY for the acknowledgement and are glad our rebuttal addressed some of your concerns. We truly appreciate your time and for maintaining the score. We will incorporate your feedback to strengthen the paper.

---

### Official Review · Reviewer_YF6x · 2026-03-11

**Soundness:** 2
**Presentation:** 3
**Significance:** 2
**Originality:** 3
**Overall Recommendation:** 3
**Confidence:** 4

**Summary:**

This paper introduces xMAE, a self-supervised pretraining framework for PPG representations that leverages paired ECG-PPG data. The core idea is that ECG captures electrical cardiac activation that temporally precedes the peripheral pulse measured by PPG, so reconstructing masked ECG from fully-visible PPG (via directional cross-attention) should encourage the PPG encoder to internalize physiologically meaningful timing structure. The framework uses continuous ECG masking with a curriculum schedule (80% to 90%) and is pretrained on ~9.4k hours of paired ICU data from MIMIC-III. The learned PPG encoder is evaluated via linear probing on 19 downstream tasks across 6 datasets spanning cardiovascular conditions, sleep staging, lab tests, and demographics, where xMAE outperforms baselines on 15/19 tasks.

**Compliance With Llm Reviewing Policy:**

Affirmed.

**Key Questions For Authors:**

1. Can you disentangle timing from morphology in what xMAE learns? The physiological grounding analysis (Figures 3-4) only examines timing features (R-peak delay, HRV). Can you show whether reconstructed ECG preserves morphological features like QRS width, P-R interval, or ST-segment characteristics? If the model only captures timing, how does this differ from simply extracting HR/HRV from PPG, which requires no pretraining at all? Demonstrating rich morphological capture would substantially strengthen the paper; the absence of it would suggest the inductive bias is less powerful than claimed.

2. How do you account for the fingertip-to-wrist PPG distribution shift? The ECG-PPG timing relationship (pulse arrival time) differs between the fingertip and the wrist due to vascular path differences. If the core benefit of xMAE is learning this timing structure during pretraining on fingertip data, why should it transfer to wrist data where the timing is different? A matched-domain evaluation (same sensor location and patient population for pretraining and evaluation) would isolate the inductive bias contribution from distribution shift confounds and could meaningfully change my assessment.

3. What happens with a classical feature-engineering baseline? Given that the analysis focuses on timing, a natural comparison is extracting standard HR, HRV (time-domain and frequency-domain), and pulse morphology features from PPG and using those directly for downstream tasks. This would calibrate whether xMAE captures something beyond what is already accessible from standard PPG signal processing.

4. Can you show the data efficiency analysis (Figure 6) across a broader set of tasks? The current version covers only ectopic beats and PVC. This is the most interesting result in the paper and expanding it would strengthen the argument that the inductive bias provides genuine benefit, particularly in low-data regimes.

**Limitations:**

The authors discuss the requirement for paired ECG-PPG data and the simplicity of MSE loss, which is reasonable. However, two important limitations are not discussed: (1) the significant distribution shift between ICU pretraining data and wearable evaluation data, particularly its implications for the timing-based learning that is the paper's central claim, and (2) the heavily skewed demographics of the pretraining population (Figure 8: ~70% White, ICU patients), which limits generalizability. The ethics section mentions the need for bias evaluation but the paper does not analyze performance across demographic subgroups.

**Strengths And Weaknesses:**

Strengths:
- The paper correctly identifies that ECG and PPG have a directional temporal relationship (electrical activation precedes mechanical pulse), and that standard multimodal MAE objectives can be satisfied via within-modality interpolation without learning cross-modal structure. The theoretical analysis in Appendix F formalizing why MM-MAE doesn't require modeling the delay while xMAE does is a nice contribution, even if the assumptions are idealized. This analysis deserves more prominence in the main text.
- 19 tasks across 6 datasets covering cardiovascular, metabolic, sleep, and demographic endpoints is thorough. The inclusion of both lab-controlled and free-living evaluation settings is good practice. The ablation study (Table 4, Figure 5) cleanly isolates the contributions of continuous masking, curriculum scheduling, and cross-reconstruction.
- The data efficiency results (Figure 6) are the most interesting results in the paper. Strong performance with only 3% of pretraining data and in few-shot regimes suggests the inductive bias is doing real work rather than acting as a regularizer at scale. This story is underemphasized relative to the aggregate benchmark numbers and could be the central selling point if expanded across more tasks.
- Code release, detailed architecture specs, unified training protocols across baselines, and thorough appendices demonstrate care for reproducibility.


Weaknesses:
- The central inductive bias is not convincingly justified. The paper's premise is that reconstructing ECG from PPG is a good pretraining task for PPG representations, but ECG and PPG share timing information (HR, HRV, pulse arrival time) while also containing substantial independent information. ECG captures electrical conduction properties (P-wave morphology, QRS axis, ST changes, conduction abnormalities) with no analog in PPG, while PPG captures peripheral vascular properties (arterial stiffness, vasomotor tone) absent from ECG. Why should forcing a PPG encoder to reconstruct electrical morphology it cannot observe produce better PPG representations than large-scale unimodal pretraining focused on what PPG actually captures? The paper's own analysis (Figures 3-4) focuses almost entirely on timing metrics (R-peak delay, HRV), which suggests the model may primarily be learning heart rate and HRV, both trivially extractable from PPG. There is no analysis of whether meaningful ECG morphological features (QRS width, ST elevation, P-R interval, T-wave shape) are preserved in the reconstruction. The authors acknowledge in the limitations that MSE loss "may not model finer-grained ECG timing information, such as P-R intervals," but this gap deserves direct investigation rather than deferral to future work, since it is central to the paper's claims.
- There is a massive distribution shift between pretraining and evaluation that is not adequately addressed. Pretraining uses MIMIC-III ICU data: critically ill patients, fingertip pulse oximetry, clinical-grade equipment. Evaluation uses wrist-worn smartwatch PPG from generally healthy ambulatory populations. This involves patient population shift (ICU vs. free-living), sensor location shift (fingertip vs. wrist), device shift (clinical vs. consumer), and signal quality shift (stationary ICU vs. motion-artifact-prone wearable). The paper frames cross-device generalization as a strength (Section 6), but the ECG-PPG timing relationship (pulse transit time) differs fundamentally between the fingertip and the wrist due to vascular path length. If the core claimed benefit is learning this timing structure, the domain gap in timing characteristics should be a central concern. A matched-domain evaluation where pretraining and evaluation PPG come from the same body location and population would help isolate the contribution of the inductive bias from confounds.
- Table 1 conflates training methodology with training scale in a way that muddies the conclusions. The baselines (Apple, LSM, DINO, SimCLR, etc.) are re-implementations trained on the authors' ~9.4k hours of MIMIC-III data. But the value proposition of methods like Apple or LSM comes from training on orders of magnitude more proprietary data. Showing that xMAE's training recipe works better on 9.4k hours of ICU data does not demonstrate that the inductive bias is generally superior. It may simply be that these other methods need more data or are not well-suited to ICU data specifically. The open-weight comparison (Table 3) is more informative but still conflates architecture, data scale, and training objective differences.
- Several evaluation datasets are very small. A1C has only 19 subjects, Hypertension (lab) has 64 subjects (1320 segments), BMI/Age (lab) has 63 subjects. With 5-fold CV, some test folds contain ~3-4 subjects for A1C. The large standard deviations (A1C: ±12.5, Hemoglobin: ±16.0, Platelets: ±16.5) reflect this instability. Claims of statistical significance via paired t-tests across 5 folds with such variance and small sample sizes should be interpreted very cautiously.
- Critical experimental details are missing from the main paper. Dataset characteristics, task definitions, sample sizes, and label distributions are almost entirely in the appendix. A reader cannot assess validity of the claims without consulting Appendix D extensively. The main paper needs at minimum a summary table of evaluation datasets.

The data efficiency results and the theoretical analysis (Appendix F) are the strongest parts of the paper. A revision that centers the narrative on these, adds morphological analysis of the reconstruction, includes a matched-domain evaluation, and reframes the contribution more precisely as learning cardiovascular timing features via ECG scaffolding could address most of these concerns.

---

> ### Author Rebuttal · Authors · 2026-03-29
>
> We thank Reviewer YF6x for the insightful comments. We are encouraged that the reviewer recognized our theoritical contributions, as well as the thoroughness of our evaluation and adherence to good practices. We address main comments below, and will include them in the revised manuscript.
>
> > Can you disentangle...?
>
> We performed feature analysis on ground-truth ECG and xMAE-reconstructed ECG. We examined five features as the reviewer suggested. **We kindly ask the reviewer to find the result table under Reviewer 3VLY.** The results show that xMAE preserves partial intra-beat interval structures, including QRS width, and PR interval, with small errors. These quantities are not captured by HR/HRV.
>
> > Why better than unimodal PPG...?
>
> Compared to unimodal PPG pretraining, our goal is not to reconstruct all ECG morphology. Instead, xMAE uses ECG as a structured supervisory signal to guide the model toward physiologically inferable components, aligning representations with the underlying electromechanical process rather than purely statistical patterns.
>
>
> > Fingertip-to-wrist shift
>
> We agree with the reviewer and conduct a match-domain comparison on wrist PPG tasks:
>
> - xMAE (Finger): pretrained on ECG + fingertip PPG
> - xMAE (Wrist): pretrained on ECG + wrist PPG
>
> We ensure comparable pretraining data scale to isolate domain effects rather than dataset size.
>
>
> | Task                        | xMAE (Finger)   | xMAE (Wrist)    |
> |-----------------------------|-----------------|-----------------|
> | Hyptn (lab)                 | 68.8 (±4.8)     | 65.9 (±8.6)     |
> | Hyptn (free-living)         | 58.5 (±1.1)     | 56.7 (±0.9)     |
> | Ectopic Beats               | 87.8 (±2.3)     | 87.4 (±2.7)     |
> | PVC                         | 81.4 (±5.1)     | 80.5 (±3.6)     |
> | Wake                        | 66.4 (±2.3)     | 65.9 (±2.4)     |
> | Light                       | 57.5 (±1.3)     | 57.4 (±1.6)     |
> | Deep                        | 55.9 (±5.3)     | 54.5 (±6.2)     |
> | REM                         | 54.5 (±2.5)     | 54.4 (±2.5)     |
>
> Despite differences in pulse arrival time, performance is similar across all tasks (AUROC, higher is better). This suggests xMAE captures transferable physiological structure, and the fingertip-to-wrist shift has only modest impact. Fingertip signals may be cleaner, contributing to strong transfer, but matched-domain pretraining does not change the overall conclusion.
>
>
> > feature-engineering baseline
>
> To address this, we extracted 35 handcrafted PPG features (**[full list](https://anonymous.4open.science/r/xMAE-icml/rebuttal/PPG_feat_table.md)**), including amplitude, morphology, derivative, and timing features. We trained a logistic regression classifier with median imputation, feature normalization, and class balancing.
>
> | Task                  | All PPG Features | PPG Timing-Only Features | xMAE               |
> |-----------------------|--------------------|---------------------------|--------------------|
> | Hyptn (lab)           | 59.7 (±13.8)       | 53.1 (±10.6)              | 68.8 (±4.8)        |
> | Hyptn (free-living)   | 58.4 (±1.7)        | 57.7 (±1.2)               | 58.5 (±1.1)        |
> | PVC                   | 76.7 (±4.9)        | 72.2 (±3.6)               | 81.4 (±5.1)        |
> | Ectopic Beats         | 83.5 (±2.2)        | 83.8 (±1.3)               | 87.8 (±2.3)        |
> | A1C                   | 58.9 (±21.3)       | 35.3 (±12.1)              | 65.1 (±12.5)       |
> | Wake                  | 63.7 (±1.2)        | 62.6 (±1.9)               | 66.4 (±2.3)        |
>
> While gaps are smaller on some tasks, indicating classical features remain competitive, xMAE overall captures information beyond handcrafted PPG features.
>
>
> > data efficiency analysis
>
> We include more tasks, and evaluate performance under 3% pretraining volume **(30% and 100% are [here](https://anonymous.4open.science/r/xMAE-icml/rebuttal/more_data_eff.md))**.
>
> | Model    | Hyptn (lab)   | Hyptn (free-living) | A1C             | Wake            | PVC             |
> |----------|-----------------|-----------------|-----------------|-----------------|-----------------|
> | xMAE     | 60.5 (±4.4)     | 55.9 (±1.2)     | 61.9 (±14.1)    | 63.6 (±1.3)     | 75.0 (±4.9)     |
> | Apple-M  | 52.9 (±3.8)     | 55.4 (±1.0)     | 47.3 (±7.8)     | 64.5 (±1.2)     | 66.1 (±4.3)     |
> | LSM      | 51.7 (±7.2)     | 54.8 (±0.7)     | 45.9 (±9.0)     | 64.9 (±1.3)     | 69.0 (±3.3)     |
>
> The results show that xMAE’s advantage is broad. At 3% pretraining volume, it outperforms baselines on most tasks, supporting that its inductive bias benefits low-data regimes and generalizes across tasks.
>
>
> > Others
>
> Due to space limit, discussions on limitations are omitted but will be clarified in the manuscript.
>
> ---
> We sincerely thank Reviewer YF6x for the thoughtful feedback and for engaging with our work. If the clarifications are helpful, we would greatly appreciate reconsideration of the score, and are happy to address any further questions.

---

> > ### Author Rebuttal · Reviewer_YF6x · 2026-04-03
> >
> > We thank the authors for the thorough rebuttal and additional experiments, which address several of our concerns.
> >
> > The matched-domain experiment (fingertip vs. wrist pretraining) is the most informative new result and substantially addresses our concern about distribution shift. The finding that fingertip-pretrained models transfer well to wrist-based evaluation tasks, with only modest degradation compared to matched-domain pretraining, suggests the learned representations capture structure that generalizes across sensor locations. We appreciate the effort to run this controlled comparison.
> >
> > The expanded data efficiency analysis (Q4) across 5 tasks at 3% pretraining volume strengthens the claim that the inductive bias provides genuine benefit in low-data regimes. The feature-engineering comparison (Q3) confirms xMAE outperforms standard handcrafted features on most tasks, though given the small evaluation datasets involved (e.g., A1C: 19 subjects), many of these differences are likely not statistically significant even where the point estimates favor xMAE. Significance testing would help confirm this either way, and we continue to believe this is important given the variance observed across folds.
> >
> > Regarding morphological features (Q1): we appreciate the authors providing the analysis from Reviewer 3VLY's response. The results are informative and partially address our concern. QRS width (median error 10ms) and PR interval (median error 0ms) show reasonable preservation, while ST-level and ST-slope show larger deviations. The authors' interpretation that ST features "reflect ventricular repolarization, which has weak or no direct analog in PPG" is fair and actually supports our original framing: xMAE captures features that have a physiological coupling pathway between ECG and PPG (timing, intervals, conduction-related structure) but not features that are purely electrical with no PPG analog. This is a more precise and defensible claim than "physiologically meaningful structure" broadly, and we would encourage the authors to adopt this framing in the revision.
> >
> > Several concerns from our original review remain unaddressed: the small evaluation dataset sizes and statistical reliability of the reported differences, the conflation of training methodology with training scale in Table 1, and the demographic skew of the pretraining population. We understand these are difficult to fully resolve in a rebuttal.
> >
> > Given the new evidence, particularly the matched-domain experiment and expanded data efficiency results, we are willing to adjust our score upward. The rebuttal demonstrates genuine engagement with the feedback and provides useful new experiments. We would encourage the authors to adopt the more precise framing of what xMAE learns (cross-modal timing and conduction-related structure rather than broad "physiologically meaningful" representations) in the revision, as we believe this is both more accurate and ultimately a stronger, more defensible contribution.

---

> > > ### Author Response · Authors · 2026-04-04
> > >
> > > We thank Reviewer YF6x for the acknowledgment. We appreciate your recognition of our additional experiments and clarifications in the rebuttal, and are especially grateful for your willingness to adjust the score upward.
> > >
> > > We also appreciate your suggestion on refining the framing of our contributions, and will incorporate this, along with your other feedback, to more accurately present our work and its relevance to this increasingly important and active research area.

---

### Official Review · Reviewer_a1K7 · 2026-03-12

**Soundness:** 3
**Presentation:** 3
**Significance:** 2
**Originality:** 3
**Overall Recommendation:** 4
**Confidence:** 3

**Summary:**

In this work, the author proposes asymmetric masking and directional cross-attention to bias learning toward an ECG-PPG temporal transition structure that reflects the underlying cardiovascular dynamics relevant to health tasks. The author applies this approach to 3.4 million ECG-PPG paired data sampled at 100 Hz to train the xMAE model. The author evaluates the xMAE on a wide range of downstream tasks related to PPG signals and finds that asymmetric masking can achieve competitive performance for cardiovascular tasks with even PPG. The author performs a physiological grounding analysis showing that the learned output closely matches the biosignal physiology.

**Compliance With Llm Reviewing Policy:**

Affirmed.

**Key Questions For Authors:**

1. The author compares most results with linear probing, and it is unclear why other fine-tuning approaches are not considered.

2. For the Case Study: Physiological Fidelity of Reconstructed ECG. I don't think one can conclude that results demonstrate that xMAE encodes meaningful ECG dynamics in its latent PPG space. The results only show the quality of ECG reconstruction but do not imply anything about encoding information into PPG space.

3. The study only shows the results with the PPG encoder, but what about the ECG encoder in terms of downstream tasks?

4. I think another naive baseline should be unimodal training with mask reconstruction on the same modality, and probe the model with such a setting

**Limitations:**

1. Since the paper only performs experiments on ECG and PPG pairs, I think it is unfair to hinting in the title as "Biosignal Representation Learning", which such an approach might not work for other signals like EEG, EMG, or breathing signals.

2. The paper only shows the downstream prediction using the PPG signals, and it is unclear whether it generalizes to other signals.

**Strengths And Weaknesses:**

Strengths

1. The author uses interesting modeling tricks in designing the models, such as the curriculum ECG masking strategy. These tricks have not been comprehensively discussed in prior works on biosignals.

2. The study experiments with quite a few different datasets in different contexts, which makes a good benchmark effort for future studies.

3. There is an interesting physiological grounding analysis that interprets the modeling outputs

4. The concept of imposing asymmetric rather than symmetric masking is an interesting direction for knowledge driven biosignal learning

Weakness

1. Since the paper only performs experiments on ECG and PPG pairs, I think it is unfair to hinting in the title as "Biosignal Representation Learning", which such an approach might not work for other signals like EEG, EMG, or breathing signals.

2. The author compares most results with linear probing, and it is unclear why other fine-tuning approaches are not considered.

3. The study only shows the results with the PPG encoder, but what about the ECG encoder in terms of downstream tasks?

4. I think another naive baseline should be unimodal training with mask reconstruction on the same modality, and probe the model with such a setting

---

> ### Author Rebuttal · Authors · 2026-03-29
>
> We thank Reviewer a1K7 for the insightful and constructive feedback. We are encouraged that the reviewer recognized our contributions. In particular, we appreciate the recognition of our asymmetric masking strategy as an interesting direction for knowledge-driven biosignal learning. We are also pleased that the reviewer found our extensive benchmarking across diverse datasets and our physiological grounding analysis to be valuable contributions to the field. We address the reviewer’s specific comments below.
>
> > Title scope (“Biosignal Representation Learning”)
>
> We thank the reviewer for pointing this out. We agree that, since our experiments are validated on ECG–PPG pairs, it is not fair to broadly claim general biosignal representation learning. We will revise the title and manuscript to more accurately reflect the scope of this work. We also appreciate the reviewer’s recognition that asymmetric masking is an interesting direction. We believe this idea captures a general principle of leveraging asymmetric observability across modalities, and exploring its applicability to other biosignal pairs (e.g., EEG, EMG, respiration, motion) is an important direction for future work.
>
> > Linear probing vs fine-tuning
>
> Thank you for this observation. Our primary goal is to evaluate the quality of the learned representations independent of task-specific adaptation, which is why we emphasize linear probing.
>
> We agree that fine-tuning provides complementary insights. In addition to the results included in the paper, we present expanded fine-tuning results below.
>
>
>
> ##### Classification (AUROC; Higher is better)
> | Task                     | xMAE-FT         | xMAE-LP             |
> |--------------------------|-----------------|------------------|
> | Hypertension (lab)       | 67.0 (±9.7)     | 68.8 (±4.8)      |
> | Hypertension (free)      | 61.4 (±0.9)     | 58.5 (±1.1)      |
> | PVC                      | 88.8 (±3.7)     | 81.4 (±5.1)      |
> | Ectopic Beats            | 93.8 (±1.1)     | 87.8 (±2.3)      |
> | A1C                      | 60.5 (±15.1)     | 65.1 (±12.5)     |
> | Hemoglobin               | 60.5 (±18.7)     | 62.0 (±16.0)     |
> | Platelets                | 69.0 (±15.8)     | 68.6 (±16.5)     |
> | Sodium                   | 63.7 (±17.2)    | 61.7 (±16.1)     |
> | Wake                     | 71.3 (±1.7)     | 66.4 (±2.3)      |
> | Light                    | 58.5 (±0.9)     | 57.5 (±1.3)      |
> | Deep                     | 60.7 (±3.2)     | 55.9 (±5.3)      |
> | REM                      | 55.3 (±3.2)     | 54.5 (±2.5)      |
>
> xMAE-FT and xMAE-LP denote fine-tuned and linear-probed results, respectively.
>
> From these results, we observe that fine-tuning generally improves performance on several tasks (e.g., PVC, ectopic beats, and sleep staging), while linear probing remains competitive on others (e.g., hypertension and some lab-related tasks). This suggests that the learned representation is informative even without task-specific adaptation, while still benefiting from fine-tuning when additional supervision is available. We also note that performance varies across tasks, partly due to differences in effective data scale. Some tasks have fewer segments per subject, which reduces the amount of training signal and leads to a higher variance.
>
> Overall, we view linear probing and fine-tuning as complementary: linear probing evaluates representation quality in a controlled setting, while fine-tuning reflects the best achievable task performance. We will include these fine-tuning results in the revised manuscript for a more comprehensive evaluation.
>
>
> > Why focus on the PPG encoder
>
> This is a valuable question. We focus on the PPG encoder because PPG is a passive sensing modality that enables continuous, unobtrusive monitoring, while ECG typically requires active user interaction (e.g., contact-based measurements on a smartwatch). Our goal is to improve representations for real-world wearable deployment, where PPG is the primary modality. Therefore, we prioritize strengthening the PPG encoder. At the same time, ECG is used as a supervisory signal during pretraining, and we evaluate reconstruction quality by comparing the reconstructed ECG against the ground truth, demonstrating that xMAE captures meaningful physiological structure.
>
> > Unimodal masked reconstruction baseline
>
> We thank the reviewer for this suggestion, and we should have made this clear. We do include this baseline in our experiments as MAE-1D (Table 1 and Table 2), which corresponds to unimodal masked reconstruction on PPG. As shown in the results, xMAE consistently outperforms this baseline across most tasks. We will revise the manuscript to more clearly highlight this comparison and avoid confusion.
>
>
> ---
> We sincerely thank Reviewer a1K7 for the thoughtful feedback and for taking the time to engage with our work. If the clarifications above are helpful, we would greatly appreciate reconsideration of the score. We are also happy to address any further questions.

---

> > ### Author Rebuttal · Reviewer_a1K7 · 2026-04-02
> >
> > The author has answered most questions, and I will keep my score.

---

> > > ### Author Response · Authors · 2026-04-04
> > >
> > > We thank Reviewer a1K7 for the acknowledgement and are glad our rebuttal addressed your concerns. We truly appreciate your time and consideration and will incorporate your suggestions to strengthen the paper.

---

### Decision · Program_Chairs · 2026-04-30

**Decision:**

Accept (regular)

**Comment:**

This paper proposes a self-supervised pretraining framework that leverages temporal relationships between ECG and PPG signals for biosignal representation learning. The paper received 2 Weak Accepts, 1 Accept, and 1 Weak Reject. The rebuttal addressed several key concerns raised by reviewers, including clarifications and new experiments. Some concerns remain e.g., small dataset sizes and framing of contributions, but the core technical contributions and empirical findings were strengthened. Overall, I recommend Accept.